# Learning Semi-Structured Sparsity for LLMs via Shared and Context-Aware Hypernetwork

**Lu Sun**[1,3*]   **Jun Sakuma**[1,2]

[1]Center for Advanced Intelligence Project (AIP), RIKEN, Tokyo 103-0027, Japan
[2]School of Computing, Institute of Science Tokyo, Tokyo 152-8550, Japan
[3]Unprecedented-scale Data Analytics Center, Tohoku University, Sendai 980-0845, Japan
`lu.sun.e4@tohoku.ac.jp, sakuma@c.titech.ac.jp`

## Abstract

Large Language Models (LLMs) achieve state-of-the-art performance but are costly to deploy in resource-constrained environments. Pruning with $n : m$ semi-structured sparsity reduces computation and enables hardware acceleration, yet existing methods face a trade-off: one-shot approaches are efficient but heuristic, while optimization-based methods are accurate but expensive. We introduce **HyperPrune**, a resource-efficient framework that directly optimizes $n : m$ sparsity. A lightweight hypernetwork, shared across layers and conditioned on learnable embeddings, generates structured masks in a one-shot, layer-wise manner. *Continual pruning* preserves cross-layer knowledge, and *feature outlier regularization* retains critical activations, unifying the strengths of heuristic and optimization-based methods. Experiments on LLaMA-7B to 70B show state-of-the-art accuracy–sparsity trade-offs on a single A100 GPU, achieving higher efficiency, accuracy, and scalability than prior approaches. HyperPrune offers a practical, scalable, and hardware-friendly solution for structured LLM pruning.

## 1   Introduction

Large Language Models (LLMs) have transformed artificial intelligence, advancing state-of-the-art performance in language understanding, reasoning, generation, and diverse applications (Brown et al., 2020; OpenAI, 2023; Grattafiori et al., 2024). Models such as GPT-4 (OpenAI, 2023), LLaMA (Touvron et al., 2023a), and OPT (Zhang et al., 2022), with billions to trillions of parameters, excel in translation, multi-step reasoning, and creative generation. However, their massive scale hinders deployment in resource-constrained environments: for example, LLaMA-70B requires over 130 GB of GPU memory, exceeding the limits of edge devices, mobile platforms, or embedded systems (Cai et al., 2020; Frantar & Alistarh, 2023; Dettmers et al., 2023; Cheng et al., 2024). This bottleneck motivates efficient compression and acceleration techniques.

To address LLM scaling challenges, the community has developed diverse compression techniques, including quantization (Dettmers et al., 2023; Frantar et al., 2022a), knowledge distillation (Hinton et al., 2015; Jiao et al., 2020), low-rank approximation (Hu et al., 2021; Xu et al., 2024a), and pruning (Frantar et al., 2022b; Dong et al., 2024). Among these, pruning is particularly attractive: it preserves the original architecture, offers fine-grained interpretability, complements quantization, and is increasingly supported by specialized hardware. Recent NVIDIA GPUs (e.g., A100, H100) natively support $n : m$ structured sparsity, enabling up to $2\times$ throughput gains in matrix multiplications, lower memory usage, and accuracy comparable to dense models (Hubara et al., 2021; Zhou et al., 2021; Fang et al., 2024; Sun et al., 2025). Empirical evidence further reinforces pruning: over-parameterized networks often generalize as well or better when pruned (Reed, 1993; Arora et al., 2018); compressing a pretrained dense model outperforms training sparse from scratch (Zhu & Gupta, 2017); and many layers tolerate aggressive pruning without performance loss (Wang et al., 2018). These factors make pruning a practical and powerful tool for LLM compression.

---

*Work done while at RIKEN AIP. Current affiliation: Tohoku University.

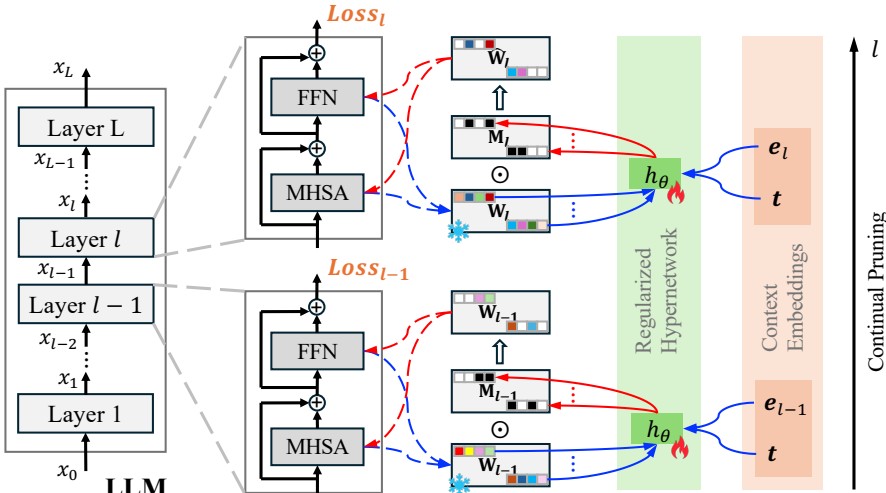

Figure 1: **The HyperPrune Framework.** HyperPrune learns semi-structured sparse patterns for LLM weight matrices, illustrated here with 2:4 sparsity. It prunes matrices $\{\mathbf{W}\}$ across components (e.g., Q, K, V, projections) in a layer-wise manner. For each layer $\ell$, a shared hypernetwork $h_\theta$ generates a binary mask $\mathbf{M}_\ell$, producing sparse weights $\widehat{\mathbf{W}}_\ell = \mathbf{W}_\ell \odot \mathbf{M}_\ell$. These are optimized via the layer loss $Loss_\ell$, which is then minimized by optimizing $\theta$. The hypernetwork is context-aware through layer ($\mathbf{e}_\ell$) and component ($\mathbf{t}$) embeddings, and further regularized by feature-outlier and continual-pruning objectives to retain important features and prevent catastrophic forgetting.

Yet pruning for LLMs remains nascent. Early iterative prune–retrain methods (Han et al., 2015; Louizos et al., 2018) are infeasible for today's billion-parameter models, prompting more efficient strategies. One-shot heuristics such as SparseGPT (Frantar & Alistarh, 2023) and Wanda (Sun et al., 2024) derive sparsity masks from small calibration sets but suffer from reconstruction loss, poor scalability, and lack of native $n : m$ support. Optimization-based approaches like MaskLLM (Fang et al., 2024) achieve higher accuracy by learning $n : m$ masks on larger calibration sets, but require massive resources (e.g., thousands of GPU hours for LLaMA-2 (Touvron et al., 2023b)). Recent methods such as MaskPro (Sun et al., 2025) reduce costs but still struggle with policy gradient variance and linearly increased memory size. Overall, current pruning approaches face two key challenges: (i) an accuracy–efficiency trade-off, where heuristics degrade at high sparsity and optimization methods are costly; and (ii) lack of efficient, direct $n : m$ support. Since hardware accelerators benefit most from $n : m$ sparsity, an effective framework must enforce such structure efficiently, within realistic budgets, without sacrificing accuracy.

In this work, we present **HyperPrune**, a resource-efficient, optimization-based framework for $n : m$ semi-structured pruning of LLMs. While heuristic methods are efficient, they cannot guarantee $n : m$ sparsity, and existing optimization-based approaches enforce it at high computational cost for billion-scale models. HyperPrune overcomes these challenges with a shared, lightweight hypernetwork $h_\theta$ that generates layer- and component-specific masks $\mathbf{M}_\ell$ conditioned on context-aware embeddings $\mathbf{e}$ and $\mathbf{t}$, producing sparse weights $\mathbf{W}_\ell \odot \mathbf{M}_\ell$, as illustrated in Fig. 1. This design enables adaptive pruning across layers and components, promotes cross-layer knowledge sharing, mitigates catastrophic forgetting, and scales efficiently to large LLMs. Crucially, we show theoretically that minimizing the reconstruction loss of pruned layers is equivalent to maximizing the mutual information between dense and pruned models under the $n:m$ constraint (Theorem 1), providing a principled relaxation of discrete mask optimization. Combined with feature-outlier and continual-pruning regularizations, HyperPrune preserves critical activations and ensures stable sequential pruning. Together, the context-aware hypernetwork, information-theoretic foundation, and novel regularizations offer an efficient, scalable, and principled approach to structured LLM pruning. Experiments demonstrate strong accuracy–sparsity trade-offs on models from 7B to 70B parameters, all on a single A100 GPU (80GB).

Our main contributions are:

- **Hypernetwork-based structured pruning.** We propose HyperPrune, the first optimization-based framework that generates $n : m$ sparse masks using a shared, context-aware hypernetwork, enabling adaptive and layer-wise structured pruning across LLM components.

- **Information-theoretic foundation for $n : m$ sparsity.** We show that maximizing mutual information under $n : m$ constraints is approximately equivalent to minimizing a reconstruction loss via a differentiable relaxation (Theorem 1), providing the first information-theoretic justification for structured mask learning.

- **Regularization for feature and knowledge preservation.** We introduce *feature outlier regularization* to preserve critical activations and *continual pruning regularization* to mitigate catastrophic forgetting during sequential layer-wise pruning.

- **Scalable and practical pruning.** HyperPrune scales to LLMs up to 70B parameters on a single A100 GPU and consistently outperforms both heuristic and optimization-based baselines on zero-shot and generation benchmarks.

## 2 RELATED WORK

**LLM Pruning.** Pruning reduces redundant parameters while aiming to preserve performance, often motivated by the Lottery Ticket Hypothesis (Frankle & Carbin, 2018; 2019). Recent LLM pruning methods fall into two main categories: **One-shot, heuristic methods.** Magnitude pruning (Han et al., 2015) removes the smallest weights and remains a strong baseline. SparseGPT (Frantar & Alistarh, 2023) and Wanda (Sun et al., 2024) prune in a single pass using layer-wise reconstruction or magnitude heuristics, reaching $50\%$ sparsity without retraining but degrading at higher levels. Extensions include OWL (Yin et al., 2024) for outlier weights, ShortGPT (Men et al., 2024) and AdaInfer (Fan et al., 2024) for layer skipping, and LaCo (Yang et al., 2024b) and SliceGPT (Ashkboos et al., 2024) for embedding compression. Pruner-Zero (Dong et al., 2024) evolves symbolic pruning metrics via genetic programming, while RIA (Zhang et al., 2024) leverages relative importance, activations, and channel permutation to outperform heuristics. **Optimization-based methods.** ALPS (Meng et al., 2024) enforces unstructured sparsity with ADMM (Boyd et al., 2011), and BESA (Xu et al., 2024b) differentiably allocates layer-wise sparsity. ShearedLLM (Xia et al., 2024) prunes efficiently at scale, while LoSA (Huang et al., 2025) integrates pruning with LoRA fine-tuning. These methods achieve higher accuracy under extreme unstructured sparsity but require thousands of GPU hours on 7B–70B models and do not natively support $n : m$ structured sparsity. MaskLLM (Fang et al., 2024) enforces strict $n : m$ patterns via probabilistic masking but incurs prohibitive memory and computational overhead due to its large mask parameters. MaskPro (Sun et al., 2025) mitigates some of this cost with a linear-space probabilistic formulation, yet it still suffers from high-variance gradients and memory usage that grows linearly with model size. In contrast, heuristic methods are efficient but brittle, while existing optimization-based approaches are accurate yet resource-intensive and often cannot scale to large LLMs with $n : m$ sparsity. Our proposed **HyperPrune** overcomes these limitations through *layer-wise optimization with a shared, context-aware hypernetwork*, whose memory footprint depends only on individual layer size rather than the full model. This design reduces both memory and compute demands while retaining high pruning quality and scalability.

**Hypernetwork-Based Methods.** Hypernetworks (Ha et al., 2017; Chauhan et al., 2024) generate parameters for a target network, enabling continual learning (Wang et al., 2024), few-shot adaptation (Wang et al., 2020), and multi-task modeling (Zhang & Yang, 2022). By conditioning on embeddings, they can produce task-specific weights, masks, or adapters. Examples include HyperShot (Sendera et al., 2023) and HyperTransformer (Karimi Mahabadi et al., 2021) for few-shot learning, and Polyhistor (Liu et al., 2022) for multi-task vision–language tasks. Continual learning approaches (von Oswald et al., 2020) use low-dimensional task embeddings to compress knowledge and transfer across tasks. Hypernetworks have also been used for compression: LoRA (Hu et al., 2021) and Text-to-LoRA (Charakorn et al., 2025) generate low-rank adapters, HyperMask (Ksiazek & Spurek, 2025) learns binary masks, HyperLogic (Yang et al., 2024a) generates logical rules, and Bayesian (Krueger et al., 2017) or noise-modulating (Eyring et al., 2025) hypernetworks improve robustness or efficiency. Despite their versatility, hypernetworks have not been applied to LLM pruning due to the enormous output space, structured sparsity requirements, and heterogeneous lay-

ers. **HyperPrune** overcomes these challenges by generating $n : m$ masks for each group of $m$ consecutive weights using a shared hypernetwork, reducing the effective output space from the full weight matrix to size $m$, while conditioning on layer- and component-specific embeddings to handle heterogeneous layers efficiently.

## 3 PRELIMINARIES

Consider a LLM parameterized by $\mathbf{W}$. The goal of pruning is to learn a binary mask $\mathbf{M}$ of the same dimension, where $M_{ij} = 0$ removes the corresponding parameter and $M_{ij} = 1$ keeps it. Given an input $\mathbf{x}$ from a calibration set $\mathcal{D}$, the standard pruning objective is:

$$\min_{\mathbf{M}} \ \mathbb{E}_{\mathbf{x} \sim \mathcal{D}} \left[ \mathcal{L}_{\text{LLM}}(\mathbf{x}; \mathbf{W} \odot \mathbf{M}) \right], \quad \text{s.t. } \|\mathbf{M}\|_0 < C, \tag{1}$$

where $\mathcal{L}_{\text{LLM}}$ is the LLM loss and the $\ell_0$ constraint enforces sparsity and $\odot$ denotes the element-wise product. Despite its simplicity, Eq. 1 is intractable at LLM scale due to the combinatorial $\ell_0$ constraint, and it yields unstructured sparsity that offers little acceleration on modern GPUs (Sun et al., 2024; Fang et al., 2024). Two main directions address these issues: **heuristic one-shot pruning** and **optimization-based pruning**.

**One-Shot, Heuristic Pruning.** Instead of learning a global mask, one-shot methods prune by directly optimizing a layer-specific mask $\mathbf{M}_\ell$ to minimize the reconstruction error:

$$\min_{\mathbf{M}_\ell} \ \mathbb{E}_{\mathbf{x} \sim \mathcal{D}} \left[ \left\| \mathbf{W}_\ell \mathbf{x}_{\ell-1} - (\mathbf{W}_\ell \odot \mathbf{M}_\ell) \mathbf{x}_{\ell-1} \right\|^2 \right], \quad \text{s.t. } \|\mathbf{M}_\ell\|_0 < C, \tag{2}$$

where $\mathbf{x}_{\ell-1}$ denotes the input to the $\ell$-th layer.[1] Various score-based heuristics (Frantar & Alistarh, 2023; Sun et al., 2024) approximate this NP-hard problem using a small calibration set. However, the resulting masks are usually *unstructured*, incompatible with hardware-friendly $n{:}m$ sparsity. Enforcing group-wise sparsity heuristically approximates $n{:}m$ but often incurs accuracy loss.

**Optimization-based Pruning.** Optimization-based methods enforce $n{:}m$ semi-structured sparsity by directly constraining each group of $m$ consecutive weights to retain exactly $n$ nonzero entries. Reformulating Eq. 1, the objective becomes

$$\min_{\{\mathbf{M}_i | \mathbf{M}_i \in \mathcal{S}^{n:m}\}} \ \mathbb{E}_{\mathbf{x} \sim \mathcal{D}} \left[ \mathcal{L}_{\text{LLM}}(\mathbf{x}; \{\mathbf{W}_i \odot \mathbf{M}_i\}) \right], \tag{3}$$

where a weight matrix $\mathbf{W} \in \mathbb{R}^{d_1 \times d_2}$ (superscript $c$ and subscript $\ell$ omitted for clarity) is partitioned row-wise into $\frac{d_1 d_2}{m}$ disjoint groups of $m$ consecutive elements. Each group $\mathbf{W}_i$ is paired with a binary mask $\mathbf{M}_i$, and the collection $\{\mathbf{M}_i | \mathbf{M}_i \in \mathcal{S}^{n:m}\}_{i=1}^{d_1 d_2 / m}$ specifies masks for all groups with constraints $\mathbf{M}_i \in \mathcal{S}^{n:m}, \forall i$. Here, $\mathcal{S}^{n:m}$ denotes the set of valid $n{:}m$ sparsity patterns. For example, in the widely adopted 2:4 case, the feasible mask set is

$$\mathcal{S}^{2:4} = \{\mathbf{s}_1, \mathbf{s}_2, \ldots, \mathbf{s}_6\} = \{[1,1,0,0], [1,0,1,0], [1,0,0,1], [0,1,1,0], [0,1,0,1], [0,0,1,1]\}, \tag{4}$$

In general, $|\mathcal{S}^{n:m}| = \binom{m}{n}$ (e.g., $|\mathcal{S}^{2:4}| = 6$, $|\mathcal{S}^{4:8}| = 70$). To make the discrete optimization in Eq. 4 tractable, Fang et al. (2024); Sun et al. (2025) sample masks from a relaxed categorical distribution over $\mathcal{S}^{n:m}$ and apply the Gumbel-Softmax trick (Jang et al., 2017; Maddison et al., 2017) for differentiability, enabling more effective scaling to large calibration sets $\mathcal{D}$ than heuristics.

## 4 METHODOLOGY

Heuristic pruning is efficient but cannot guarantee $n : m$ structured masks, while optimization-based methods enforce $n : m$ sparsity but are costly for billion-scale models. We propose **HyperPrune**, a framework for efficient structured pruning of LLMs. HyperPrune uses a lightweight shared hypernetwork with contextual embeddings to generate $n : m$ masks sequentially, incorporates continual pruning to preserve cross-layer knowledge, and applies feature-outlier regularization to retain critical weights. We further enable end-to-end optimization via continuous relaxation and improve efficiency and generalization through sparse-prior transfer.

---

[1]Strictly speaking, a superscript $c$ should be used for both $\mathbf{x}_{\ell-1}$ and $\mathbf{W}_\ell$ to indicate the component $c \in \mathcal{C}$. Here, $\mathcal{C} = \{q, k, v, o, u, d, g\}$ refers to the set of Transformer subcomponents, where $q, k, v, o$ correspond to multi-head self-attention (MHSA) and $u, d, g$ correspond to the feed-forward network (FFN). For readability, we omit these superscripts throughout the paper.

## 4.1 Efficient Layer-wise Mask Optimization via Hypernetwork

**Layer-wise Mask Optimization.** Unlike one-shot pruning in Eq. 2 that relies on heuristic scores or global methods in Eq. 3 that optimize masks for the entire model, we adopt a layer-wise strategy. For each Transformer layer $\ell \in [L]$ with function $f$, the objective is:

$$\min_{\{\mathbf{M}_{\ell,i} | \mathbf{M}_{\ell,i} \in \mathcal{S}^{n:m}\}} \mathbb{E}_{\mathbf{x} \sim \mathcal{D}} \left[ \|f(\{\mathbf{W}_{\ell,i}\}, \mathbf{x}_{\ell-1}) - f(\{\mathbf{W}_{\ell,i} \odot \mathbf{M}_{\ell,i}\}, \mathbf{x}_{\ell-1})\|^2 \right], \tag{5}$$

where $\{\mathbf{M}_{\ell,i}\}$ is the set of $n:m$ groups (defined in Eq. 3) of the $\ell$-th layer mask $\mathbf{M}_\ell$. This approach scales under resource constraints, as only the current layer's weights and activations need GPU memory, and it provides a stronger objective than one-shot pruning by optimizing at the layer rather than component level. Two challenges remain: (i) pruning layers independently may lose cross-layer knowledge, and (ii) directly optimizing large binary masks $\mathbf{M}_\ell$ is infeasible for modern LLMs—for instance, a single FFN projection in LLaMA-70B contains over 1.7B parameters, making naive layer-wise optimization impractical without additional parameterization or compression.

**Shared Hypernetwork for $n : m$ Sparsity.** To overcome the limitations of Eq. 5, rather than directly learning independent layer-wise masks $\mathbf{M}_\ell$, **HyperPrune** employs a lightweight shared hypernetwork to generate $n : m$ structured masks across all layers, enhancing both memory efficiency and cross-layer knowledge transfer. For a Transformer layer $\ell$ with weight matrix $\mathbf{W}_\ell$, and under the grouping scheme of Eq. 3, the $i$-th group mask $\mathbf{M}_{\ell,i}$ is sampled from its corresponding weight group $\mathbf{W}_{\ell,i}$ via the shared hypernetwork $h_\theta : \mathbb{R}^m \mapsto \mathbb{R}^m$:

$$\mathbf{M}_{\ell,i} \sim h_\theta(\mathbf{W}_{\ell,i}), \quad \forall i \in \left[\frac{d_1 d_2}{m}\right], \forall \ell \in [L]. \tag{6}$$

For each group $\mathbf{W}_{\ell,i}$, the hypernetwork outputs logits $\mathbf{p}_{\ell,i}$ over the valid $n:m$ mask set $\mathcal{S}^{n:m} = \{\mathbf{s}_j \in \{0,1\}^m : \|\mathbf{s}_j\|_0 = n\}$ based on $\mathbb{P}(\mathbf{M}_{\ell,i} = \mathbf{s}_j) = p_{\ell,ij}, \mathbf{s}_j \in \mathcal{S}$. The mask for group $i$ is generated contextually from its weights $\mathbf{W}_{\ell,i}$:

$$\mathbf{M}_{\ell,i} \sim \text{Categorical}(\mathbf{p}_{\ell,i}), \quad \mathbf{p}_{\ell,i} = \text{Softmax}(h_\theta(\mathbf{W}_{\ell,i})), \tag{7}$$

where $h_\theta : \mathbb{R}^m \mapsto \mathbb{R}^m$ is the shared hypernetwork and $\mathcal{S}^{n:m}$ the valid mask set defined in Eq. 4. This shared, group-wise design drastically reduces hypernetwork size while capturing local dependencies within each $m$-sized block. For example, producing 6 logits from $m=4$ weights requires only a few thousand parameters, whereas a naïve full-matrix hypernetwork for the FFN up-projection ($\mathbf{W}_u \in \mathbb{R}^{4096 \times 11008}$ in LLaMA-7B) would demand tens of millions of input/output dimensions—*billions* of parameters, several orders larger. Handling extremely high-dimensional LLM layers under strict $n:m$ sparsity makes this shared, local-mask generation a necessary and non-trivial innovation for memory- and compute-efficient structured pruning.

**Context-Aware Embeddings.** To adapt the shared hypernetwork to different pruning contexts across layers and components, we introduce trainable conditional embeddings. A **layer embedding** $\mathbf{e}_\ell \in \mathbb{R}^d$ identifies the $\ell$-th layer for depth-specific adaptation, while a **component embedding** $\mathbf{t} \in \mathbb{R}^d$ specifies the pruned component—Query ($\mathbf{t}_q$), Key ($\mathbf{t}_k$), Value ($\mathbf{t}_v$), Output ($\mathbf{t}_o$) for MHSA, and Up-projection ($\mathbf{t}_u$), Down-projection ($\mathbf{t}_d$), Gating ($\mathbf{t}_g$) for FFN. These embeddings are learned jointly with the hypernetwork, adding $d \times (L + 7)$ parameters: one per layer and seven shared globally. Initialization and optimization details are in Appendix D.4.

Using the shared hypernetwork and embeddings, HyperPrune generates context-aware masks:

$$\min_{\theta, \mathbf{e}, \mathbf{t}} \mathbb{E}_{\mathbf{x} \sim \mathcal{D}} \mathbb{E}_{\{\mathbf{M}_{\ell,i} \sim h_\theta(\mathbf{W}_{\ell,i}, \mathbf{e}_\ell, \mathbf{t})\}} \left[ \|f(\{\mathbf{W}_{\ell,i}\}, \mathbf{x}_{\ell-1}) - f(\{\mathbf{W}_{\ell,i} \odot \mathbf{M}_{\ell,i}\}, \mathbf{x}_{\ell-1})\|^2 \right]. \tag{8}$$

This formulation captures structural regularities across layers while remaining flexible to component-level variations. HyperPrune thus balances generalization (via shared parameters) and specialization (via embeddings), yielding more effective and transferable pruning.

**Information-Theoretic Interpretations.** From an information-theoretic view, optimizing Eq. 8 corresponds to maximizing the mutual information between dense and pruned models, formalized in Theorem 1.

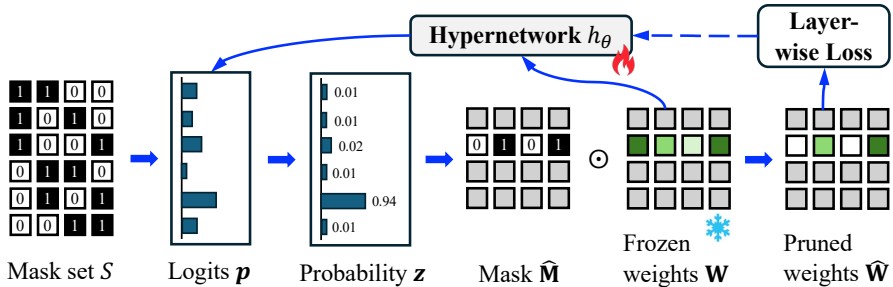

Figure 2: Illustration of generating $n:m$ semi-structured sparsity masks via continuous relaxation with a hypernetwork (example: $2:4$). The hypernetwork takes frozen weights $\mathbf{W}$ as input and outputs logits $\mathbf{p}$ over mask patterns. Probabilities $\mathbf{z}$ and the relaxed mask $\widehat{\mathbf{M}}$ are computed via Eq. 37, yielding pruned weights $\widehat{\mathbf{W}} = \mathbf{W} \odot \widehat{\mathbf{M}}$ used in Eq. 12. This relaxation allows gradients to flow through mask selection, enabling end-to-end updates of $\theta$.

**Theorem 1** (Stochastic Relaxation of $n:m$ Masked Linear Approximation). *Let $\mathbf{W} \in \mathbb{R}^{d \times d}$ and $\mathbf{x} \sim \mathcal{D}$. Partition a binary mask $\mathbf{M}$ into $B = d^2/m$ disjoint blocks $\{\mathbf{M}_i\}$ of size $m$, each constrained to satisfy $n:m$ sparsity. Let $\mathbb{E}[\mathbf{M}]$ denote the block-wise expectation of these random masks. Then,*

$$\max_{\mathbf{M} \in \mathcal{S}_{n:m}} I(f(\mathbf{W}, \mathbf{x}); f(\mathbf{W} \odot \mathbf{M}, \mathbf{x})) \;\Leftrightarrow\; \min_{\{\mathbf{p}_i\}} \mathbb{E}_{\mathbf{x} \sim \mathcal{D}} \| f(\mathbf{W}, \mathbf{x}) - f(\mathbf{W} \odot \mathbb{E}[\mathbf{M}], \mathbf{x}) \|^2,$$

*where $\{\mathbf{p}_i\}$ parameterize a differentiable relaxation of the discrete $n:m$ mask selection.*

**Proof Sketch.** Let $\mathbf{Z} = f(\mathbf{W}, \mathbf{x})$ and $\hat{\mathbf{Z}} = f(\mathbf{W} \odot \mathbf{M}, \mathbf{x})$. Under a small perturbation assumption, the sparse output can be viewed as a noisy version of the dense output, and for linear layers with Gaussian inputs, maximizing their mutual information is approximately equivalent to minimizing the expected squared difference $\mathbb{E}_{\mathbf{x}} \|\mathbf{Z} - \hat{\mathbf{Z}}\|^2$. To enable differentiable optimization under $n:m$ constraints, we introduce a block-wise categorical distribution over feasible masks and replace $\mathbf{M}$ with its expectation, which by linearity yields the surrogate objective $\mathbb{E}_{\mathbf{x}} \| f(\mathbf{W}, \mathbf{x}) - f(\mathbf{W} \odot \mathbb{E}[\mathbf{M}], \mathbf{x}) \|^2$. The full proof is provided in Appendix A.

This theorem shows that learning stochastic $n:m$ masks via soft probabilities is equivalent to maximizing the information preserved between dense and pruned layers. As a result, discrete structured sparsity can be optimized through a smooth reconstruction objective, enabling efficient end-to-end training of the hypernetwork.

### 4.2 REGULARIZED HYPERNETWORK FOR ENHANCED MASK LEARNING

**Continual Pruning Regularization.** Layer-wise pruning risks *catastrophic forgetting*: when optimizing the hypernetwork $h$ for the current layer $\ell$, knowledge from earlier layers $(1, \ldots, \ell-1)$ may be overwritten. The formulation in Eq. 8 targets only the current layer and lacks consistency with prior ones. Inspired by continual learning (Wang et al., 2024), HyperPrune preserves past knowledge by regularizing the hypernetwork to remain consistent on previously pruned layers. In this *continual pruning* setting (Mallya et al., 2018; Golkar et al., 2019; Ksiazek & Spurek, 2025), the hypernetwork parameters $\theta'$ after pruning layer $\ell-1$ initialize those for layer $\ell$. To mitigate forgetting, we penalize discrepancies between the outputs of $h_{\theta'}$ and $h_\theta$ over earlier layers:

$$\mathcal{R}_{\text{continual}} = \mathbb{E}_{\mathbf{W} \sim \mathcal{D}_W} \left[ \frac{1}{(\ell-1)} \sum_{\ell'=1}^{\ell-1} \| h_{\theta'}(\mathbf{W}_{\ell'}, \mathbf{e}_{\ell'}, \mathbf{t}) - h_\theta(\mathbf{W}_{\ell'}, \mathbf{e}_{\ell'}, \mathbf{t}) \|^2 \right]. \tag{9}$$

The factor $\frac{1}{(\ell-1)}$ normalizes across layers, preventing scale-up of the regularization. In practice, since only $\mathbf{W}_\ell$ is loaded during pruning, the expectation can be approximated by caching a small subset $\mathcal{D}_W$ of $\mathbf{W}$ from previous layers (discussed in Appendix D.8). This regularization serves as *functional knowledge distillation*, aligning past and current hypernetworks. More broadly, it can be seen as an information-theoretic constraint (e.g., KL divergence or mutual information preservation) that stabilizes behavior across layers and prevents abrupt shifts that degrade mask quality.

**Feature Outlier Regularization.** Large-scale LLMs (typically $>6B$ parameters) often produce hidden features with unusually high magnitudes. These *feature outliers* are not noise; they encode key semantic and structural information and strongly influence predictions (Dettmers et al., 2022). Naively pruning weights linked to such features can severely degrade performance, especially under structured sparsity. To mitigate this, we introduce a feature-aware regularization that favors preserving weights interacting with high-magnitude activations (large weight and input norms). Formally:

$$\mathcal{R}_{\text{outlier}} = \mathbb{E}_{\mathbf{x} \sim \mathcal{D}} \mathbb{E}_{\mathbf{M} \sim h_\theta} \left[ \left\| (\mathbf{W}_\ell \odot \mathbf{M}_\ell) \cdot \text{Diag}(\mathbf{x}_{\ell-1}) \right\|^2 \right], \tag{10}$$

where $\text{Diag}(\cdot)$ forms a diagonal matrix. This biases pruning toward directions aligned with high-magnitude activations, protecting critical semantic channels and preserving accuracy and robustness. Interestingly, for a single component (ignoring $\mathbb{E}_{\mathbf{M}}$ for simplicity), this regularization can be reformulated as

$$\mathbb{E}_{\mathbf{x} \sim \mathcal{D}} \left[ \left\| \widehat{\mathbf{W}}_\ell \cdot \text{Diag}(\mathbf{x}_{\ell-1}) \right\|^2 \right] = \sum_{i,j} \left( \widehat{W}_{\ell,ij} \cdot \mathbb{E}_{\mathbf{x} \sim \mathcal{D}}[x_{\ell-1,j}] \right)^2 + \sum_{i,j} \widehat{W}_{\ell,ij}^2 \, \text{Var}[x_{\ell-1,j}], \tag{11}$$

with $\widehat{\mathbf{W}}_\ell = \mathbf{W}_\ell \odot \mathbf{M}_\ell$. The first term recovers Wanda's importance score (Sun et al., 2024), based on average weight–activation alignment. The second term captures feature variance, ensuring even zero-mean features affecting outputs are preserved. Together, they provide a variance-aware importance measure for more robust pruning under stochastic or heterogeneous inputs.

**Formulation of HyperPrune.** By combining continual pruning regularization (9), feature outlier regularization (10), and hypernetwork-based pruning (8), the full objective of **HyperPrune** is:

$$\min_{\theta, \mathbf{e}, \mathbf{t}} \mathbb{E}_{\mathbf{x} \sim \mathcal{D}} \mathbb{E}_{\mathbf{M}} \left[ \left\| f(\mathbf{W}_\ell, \mathbf{x}_{\ell-1}) - f(\widehat{\mathbf{W}}_\ell, \mathbf{x}_{\ell-1}) \right\|^2 \right] - \lambda_1 \mathcal{R}_{\text{outlier}} + \lambda_2 \mathcal{R}_{\text{continual}}$$

$$\text{s.t.} \quad \widehat{\mathbf{W}}_{\ell,i} = \mathbf{W}_{\ell,i} \odot \mathbf{M}_{\ell,i}, \ \mathbf{M}_{\ell,i} \sim \text{Categorical}(\mathbf{p}_{\ell,i}), \ \mathbf{p}_{\ell,i} = \text{Softmax}(h_\theta(\mathbf{W}_{\ell,i}, \mathbf{e}_\ell, \mathbf{t})),$$

$$\mathbb{P}(\mathbf{M}_{\ell,i} = \mathbf{s}_j) = p_{\ell,ij}, \ \mathbf{s}_j \in \mathcal{S}, \ i \in \left[ \frac{d_1 d_2}{m} \right], \ \ell \in [L], \tag{12}$$

where $\lambda_1, \lambda_2 \geq 0$ are regularization hyperparameters.

HyperPrune is a hypernetwork-based, continual-learning-inspired framework for $n{:}m$ semi-structured pruning of LLMs. Rather than jointly optimizing masks across all layers, it prunes each layer sequentially via a shared hypernetwork, retaining knowledge from previous layers to mitigate catastrophic forgetting and reduce memory usage. Categorical mask sampling is relaxed with the Gumbel-Softmax trick for end-to-end gradient-based optimization (Algorithm 1); details are in Appendix B. Fig. 1 illustrates this context-aware, hypernetwork-based mask generation.

## 5 EXPERIMENTS

### 5.1 EXPERIMENTAL SETUP

**Models and Baselines.** We evaluate **HyperPrune**[2] on a diverse set of pre-trained large language models, including **LLaMA-2** (7B/13B/70B) (Touvron et al., 2023b), to assess both performance and scalability. We compare against representative heuristic and optimization-based pruning methods: **Magnitude** (Han et al., 2015), **Wanda** (Sun et al., 2024), **SparseGPT** (Frantar & Alistarh, 2023), **Pruner-Zero** (Dong et al., 2024), and **MaskPro** (Sun et al., 2025). We exclude MaskLLM (Fang et al., 2024) due to its prohibitive computational cost, which requires thousands of GPU hours and 520K calibration samples even for LLaMA-2-13B. Instead, we adopt MaskPro as a strong optimization-based baseline that achieves competitive results with substantially lower cost (Sun et al., 2025). Full experimental configurations and additional results are provided in Appendix D.

**Evaluation and Datasets.** For methods requiring calibration data, either for pruning-score computation or optimization, we follow SparseGPT and Wanda and use 128 sequences sampled from the C4 training set (Raffel et al., 2020), with context length matching the model. For HyperPrune, we

---

[2]https://github.com/futuresun912/HyperPrune.git

---

**Algorithm 1** HyperPrune: Semi-Structured Pruning via Shared and Context-Aware Hypernetwork

---
1: **Input:** Pretrained LLM with $L$ layers, calibration data $\mathcal{D}$
2: **Initialize:** Shared hypernetwork $h_\theta$, layer embeddings $\{\mathbf{e}_\ell\}$, task embeddings $\{\mathbf{t}_c\}$
3: **for** layer $\ell = 1$ to $L$ **do**
4: Load $\mathbf{W}_\ell$ into GPU
5: **while** not converged **do**
6: Get input activation $\mathbf{x}_{\ell-1}$ from $\mathcal{D}$
7: **for** block $i$ in $\mathbf{W}_\ell$ **do**
8: Compute logits: $\log \mathbf{p}_{\ell,i} = h_\theta(\mathbf{W}_{\ell,i}, \mathbf{e}_\ell, \mathbf{t})$
9: Sample structured mask $\widehat{\mathbf{M}}_{\ell,i}$ via Gumbel-Softmax
10: Prune weights: $\widehat{\mathbf{W}}_{\ell,i} = \mathbf{W}_{\ell,i} \odot \widehat{\mathbf{M}}_{\ell,i}$
11: **end for**
12: Forward pass: $\mathbf{x}_\ell = f(\widehat{\mathbf{W}}_\ell, \mathbf{x}_{\ell-1})$
13: Compute reconstruction loss $\mathcal{L}_{\text{recon}}$ and outlier regularization $\mathcal{R}_{\text{outlier}}$
14: Compute continual pruning regularization from past layers $\mathcal{R}_{\text{continual}}$
15: Update $\theta$, $\mathbf{e}_\ell$, and $\{\mathbf{t}\}$ using total loss $\mathcal{L} = \mathcal{L}_{\text{recon}} - \lambda_1 \cdot \mathcal{R}_{\text{outlier}} + \lambda_2 \cdot \mathcal{R}_{\text{continual}}$
16: **end while**
17: Freeze $h_\theta$ as $h_{\theta'}$ for future regularization
18: Unload $\mathbf{W}_\ell$ to free memory
19: **end for**
20:
21: **Return** Final pruned model with structured sparse weights

---

optimize the Gumbel logits for 200 steps while keeping LLM weights frozen. Consistent with prior work, C4 is used for pruning calibration and Wikitext (Merity et al., 2017) for perplexity evaluation (PPL). We further adopt the EleutherAI LM Harness (Gao et al., 2024) for zero-shot evaluation on downstream tasks (measured by accuracy), providing a standardized and reproducible benchmark. All experiments are conducted on a single NVIDIA A100 GPU (80GB).

**Sparse Pattern.** We exclude the embedding layer and the final classification head from pruning, as pruning them significantly harms performance while only accounting for about 1% of model parameters. All remaining linear layers are pruned under the 2:4 semi-structured sparsity pattern. For heuristic methods that do not natively support semi-structured sparsity, we follow the procedure in (Frantar & Alistarh, 2023; Sun et al., 2024), selecting two elements to retain within each group of four consecutive weights based on their pruning scores.

## 5.2 EXPERIMENTAL RESULTS

**Performance Evaluation.** This section evaluates the efficacy of our proposed pruning method with respect to two key questions: (i) whether generative performance is preserved after structured pruning, measured by perplexity on standard language modeling benchmarks such as Wiki-Text (Merity et al., 2017), and (ii) whether zero-shot generalization is maintained across diverse downstream tasks using the EleutherAI LM Harness (Gao et al., 2024). As shown in Table 1, our method, **HyperPrune**, consistently achieves the best trade-off between perplexity and accuracy across all model scales. For **LLaMA-2 7B**, HyperPrune reduces Wikitext perplexity to **10.11** and attains the highest mean accuracy of **53.76**, outperforming heuristic methods (e.g., SparseGPT at 51.00) and optimization-based methods (e.g., MaskPro at 52.81). On **LLaMA-2 13B**, HyperPrune leads with a mean accuracy of **59.25**, slightly surpassing MaskPro (58.97) while maintaining competitive perplexity (**7.60**). For **LLaMA-2 70B**, HyperPrune achieves a mean accuracy of **68.57** with low perplexity (**5.13**), exceeding all baselines. These results demonstrate that end-to-end training of a compact shared hypernetwork for mask generation enables HyperPrune to robustly retain generative quality and cross-task generalization. In contrast, traditional heuristics suffer from estimation errors in weight scoring, and optimization-based methods such as MaskPro face memory limitations on large models (e.g., LLaMA-2 70B) or MaskLLM require substantially more calibration samples and GPU resources than HyperPrune, highlighting its effectiveness and efficiency (See Table 8).

Table 1: Evaluation of language modeling and zero-shot task performance under 2:4 sparsity on models with frozen weights. Pruning methods were calibrated on the C4 dataset and evaluated on Wikitext-2 (perplexity) and seven zero-shot tasks (accuracy), following the methodology of (Frantar & Alistarh, 2023; Sun et al., 2024). Best results are in bold and second-best are underlined.

| Method | Wikitext ↓ | BoolQ ↑ | RTE ↑ | HellaS. ↑ | WinoG. ↑ | ARC-E ↑ | ARC-C ↑ | OBQA ↑ | Mean ↑ |
|---|---|---|---|---|---|---|---|---|---|
| **LLaMA-2 7B** | 5.12 | 77.74 | 62.82 | 57.17 | 68.90 | 76.39 | 43.52 | 31.40 | 59.71 |
| - Magnitude | 52.85 | 56.33 | 51.38 | 42.47 | 61.03 | 59.68 | 27.61 | 21.98 | 45.78 |
| - SparseGPT | 10.39 | 70.56 | 58.90 | 43.46 | **66.71** | 64.17 | 29.95 | 23.22 | 51.00 |
| - Wanda | 11.09 | 67.62 | 53.17 | 40.95 | 62.38 | 61.76 | 31.23 | 24.32 | 48.78 |
| - Pruner-Zero | 10.35 | 69.14 | 53.43 | 54.68 | 60.54 | 61.57 | 32.17 | **32.60** | 52.02 |
| - MaskPro | 12.29 | **71.12** | **61.37** | 46.18 | 65.82 | 66.12 | **32.85** | 26.20 | 52.81 |
| **HyperPrune** | **10.11** | 70.69 | 60.22 | **55.28** | 64.50 | **66.32** | 32.55 | 26.76 | **53.76** |
| **LLaMA-2 13B** | 4.57 | 80.52 | 65.34 | 60.06 | 72.22 | 79.42 | 48.46 | 35.20 | 63.03 |
| - Magnitude | 8.31 | 65.70 | 54.25 | 50.17 | 62.06 | 62.51 | 31.72 | 23.23 | 49.95 |
| - SparseGPT | 8.27 | 76.78 | 59.48 | 46.61 | 68.69 | 70.64 | 36.66 | 25.50 | 54.91 |
| - Wanda | 8.29 | 76.83 | 61.26 | 47.83 | 66.92 | 69.34 | 36.87 | 26.43 | 55.07 |
| - Pruner-Zero | **7.53** | 77.89 | 56.68 | 63.37 | 67.72 | 69.70 | 37.29 | **36.00** | 58.38 |
| - MaskPro | 8.16 | **78.05** | **62.22** | 63.18 | 67.67 | 70.93 | **37.56** | 33.18 | 58.97 |
| **HyperPrune** | 7.60 | 77.63 | 61.84 | **64.20** | 67.77 | **71.00** | 37.50 | 34.80 | **59.25** |
| **LLaMA-2 70B** | 3.12 | 83.40 | 67.87 | 66.10 | 78.06 | 82.55 | 54.44 | 37.20 | 67.08 |
| - Magnitude | 6.33 | 73.22 | 57.03 | 58.41 | 74.30 | 76.17 | 45.23 | 35.46 | 59.97 |
| - SparseGPT | 5.44 | 79.53 | **70.75** | 59.03 | 76.62 | 78.96 | 48.57 | 33.82 | 64.61 |
| - Wanda | 5.21 | **82.18** | 69.83 | 59.37 | 76.25 | 79.33 | 47.29 | 34.84 | 64.16 |
| - Pruner-Zero | **4.87** | 81.40 | 67.87 | 77.60 | 74.11 | 79.45 | **50.00** | 43.40 | 67.69 |
| **HyperPrune** | 5.13 | 81.75 | 70.20 | **77.83** | **77.60** | **79.50** | 49.62 | **43.51** | **68.57** |

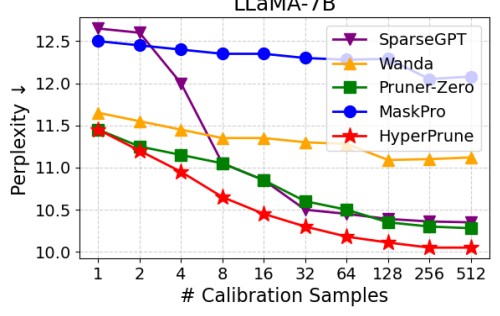

(a) Scalability on language modeling

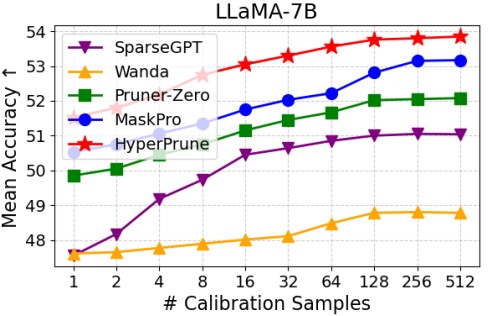

(b) Scalability on zero-shot generalization

Figure 3: Perplexity on Wikitext (a) and mean accuracy across seven zero-shot tasks (b) for different pruning methods, evaluated with increasing numbers of calibration samples (1, 2, 4, 8, 16, 32, 64, 128, 256, and 512). The results illustrate how performance scales with additional calibration data.

**Scalability Evaluation.** To evaluate whether HyperPrune can generate scalable pruning masks that exploit structural redundancy across layers and modules while minimizing performance degradation, we adopt a held-out calibration protocol following MaskLLM. As shown in Fig. 3, **Hyper-Prune** consistently outperforms competing methods in both Wikitext perplexity and zero-shot mean accuracy, with performance steadily improving as more calibration samples are used. With 512 samples, it achieves the lowest perplexity ($\approx 10.2$) and highest mean accuracy ($> 53\%$), indicating strong data efficiency and scalability. Although MaskPro performs competitively, it is consistently surpassed by HyperPrune, while heuristic approaches such as Wanda and SparseGPT show weaker scalability and limited gains from additional calibration data. These results demonstrate that HyperPrune not only delivers superior overall performance but also effectively leverages increasing calibration data, making it a robust and scalable solution for pruning large language models.

**Ablation Studies.** We evaluate the contribution of each HyperPrune component on LLaMA-2-7B by removing layer embeddings (le), component embeddings (ce), continual pruning (cp), or feature-outlier (fo) regularization. Removing either embedding consistently degrades performance (e.g.,

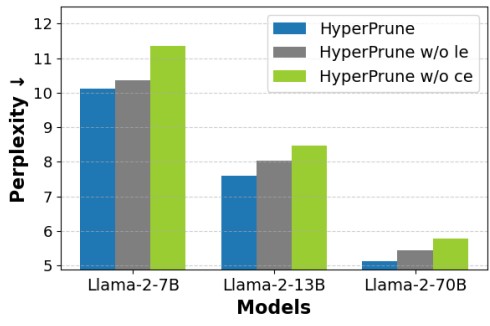
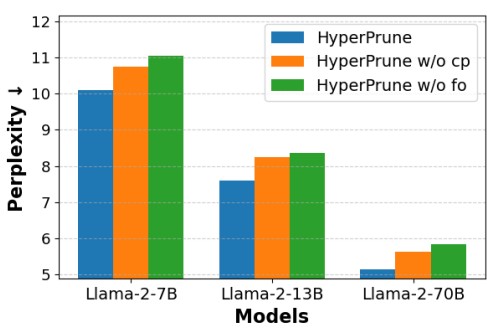

(a) Effect of context-aware embedding

(b) Effect of regularization

Figure 4: Ablation study of HyperPrune on the Wikitext dataset. (a) Impact of context-aware embeddings, where **le** and **ce** denote layer and component embeddings. (b) Impact of regularizations, where **cp** and **fo** denote continual pruning and feature outlier retention.

perplexity increases from 10.11 to 11.36), confirming that contextual conditioning is important for generating high-quality structured masks, with layer embeddings having a slightly larger impact than component embeddings. Similarly, disabling cp or fo increases perplexity (e.g., 10.11 to 11.06 without fo), highlighting their role in stabilizing sequential pruning and preserving salient features during sparsification. Overall, these ablations show that both context-aware embeddings and the proposed regularization strategies are necessary to maintain low perplexity and achieve effective layer-wise structured pruning.

**Computational and Memory Efficiency.** We evaluate the practical efficiency of HyperPrune under 2:4 structured sparsity. On an A100 GPU, kernel-level speedups of $1.55$–$1.65\times$ translate into an end-to-end latency reduction for LLaMA-2-7B from 248 ms to 174 ms ($1.43\times$; see Appendix D.9), demonstrating tangible inference gains under realistic deployment settings. Importantly, these improvements arise from hardware-aligned structured sparsity, ensuring that theoretical sparsity translates into actual runtime acceleration. HyperPrune is also highly efficient during mask optimization (Appendix D.10, D.11). Tables 8 and 9 show that it reduces pruning-mask training cost by nearly an order of magnitude, requiring only 7–15 GPU hours and 15–22 GB memory for LLaMA-2 7B/13B. This is substantially lower than MaskPro and dramatically more efficient than MaskLLM, which requires 1200–2300 GPU hours and hundreds of GB of memory, while HyperPrune remains feasible even at 70B scale. Overall, these results demonstrate that HyperPrune achieves strong computational and memory efficiency without sacrificing scalability to modern large-scale LLMs.

# 6 CONCLUSION AND FUTURE WORK

We introduced **HyperPrune**, a practical framework for learning $n : m$ semi-structured sparsity in LLMs. HyperPrune employs a compact shared hypernetwork to generate block-wise masks in a sliding, layer-wise fashion, conditioned on learned embeddings. It incorporates continual-learning and feature-outlier regularizers to preserve cross-layer knowledge and protect critical activations. This enables direct optimization for hardware-friendly $n : m$ sparsity with low memory and compute overhead, supporting models from 7B to 70B parameters on a single A100 GPU. Experiments on LLaMA show that HyperPrune maintains language modeling performance and provides tangible speedups under 2:4 sparsity, demonstrating its efficiency for LLM deployment.

Nonetheless, HyperPrune has limitations. It relies on sufficient calibration data, potentially reducing robustness under domain shift, and the lightweight hypernetwork faces a capacity–generalization trade-off, motivating adaptive or hierarchical designs. Future work includes evaluation on diverse inference stacks and hardware, and integration with complementary techniques such as quantization or low-rank adaptation to maximize practical impact.

ACKNOWLEDGMENTS

We sincerely thank the anonymous reviewers for their constructive feedback and valuable suggestions, which have helped improve the clarity and quality of this work. This work was supported in part by JST (JPMJNX25C2, JPMJKP24C3, JPMJCR23M4, JPMJCR21D3) and JSPS (23H00483, 120251002, and 26K21320).

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

## A  PROOF OF THEOREM 1: STOCHASTIC RELAXATION OF $n:m$ SPARSE MASKING VIA EXPECTED MSE LOSS

Here we present the full version of Theorem 1 along with its detailed proof, which were omitted from the main text for clarity.

**Theorem 2** (Stochastic Relaxation of $n:m$ Masked Linear Approximation). *Let $\mathbf{W} \in \mathbb{R}^{d \times d}$ and $\mathbf{x} \sim \mathcal{D}$. Partition a binary mask $\mathbf{M} \in \{0,1\}^{d \times d}$ into $B = d^2/m$ blocks $\{\mathbf{M}_i\}$ of size $m$, with*

$$\mathbf{M}_i \sim Categorical(p_i), \quad p_i \in \Delta^{|\mathcal{S}|-1}, \quad \mathcal{S}_{n:m} = \{\mathbf{s} \in \{0,1\}^m : \|\mathbf{s}\|_0 = n\}.$$

*Define the relaxed mask $\mathbb{E}[\mathbf{M}] = [\sum_j p_{1j}\mathbf{s}_j, \ldots, \sum_j p_{Bj}\mathbf{s}_j] \in [0,1]^{d \times d}$. Then*

$$\max_{\mathbf{M} \in \mathcal{S}_{n:m}} I(f(\mathbf{W}, \mathbf{x}); f(\mathbf{W} \odot \mathbf{M}, \mathbf{x})) \Leftrightarrow \min_{\{p_i\}} \mathbb{E}_{\mathbf{x} \sim \mathcal{D}} \|f(\mathbf{W}, \mathbf{x}) - f(\mathbf{W} \odot \mathbb{E}[\mathbf{M}], \mathbf{x})\|^2,$$

*where $\{p_i\}$ provide a differentiable relaxation of discrete $n:m$ sparsity and $f$ denotes MHSA/FFN layers (linear activation).*

*Proof.* We split the proof into two main parts: (i) the reduction from mutual information to a mean squared error (MSE) objective under a small-noise approximation, and (ii) the stochastic relaxation of the discrete mask into a continuous surrogate via expectation.

**Part 1: From Mutual Information to Expected MSE.** We consider the mutual information between the dense and sparse linear layer outputs: $I\big(f(\mathbf{W}, \mathbf{x}); f(\mathbf{W} \odot \mathbf{M}, \mathbf{x})\big)$, where $f$ denotes the function of MHSA or FFN with linear activation, and we omit the layer subscript $L$ for brevity. We aim to show that minimizing the MSE between the outputs of a dense model and a pruned model is equivalent to maximizing the mutual information between them under mild assumptions. Let us denote:

- $\mathbf{Z} := f(\mathbf{W}, \mathbf{x})$ — the output of the original dense layer,

- $\hat{\mathbf{Z}} := f(\mathbf{W} \odot \mathbf{M}, \mathbf{x})$ — the output of the pruned sparse layer,

- $\mathbf{M} \in \mathcal{S}_{n:m}$ — the binary sparsity mask constrained by the $n:m$ sparsity pattern.

**Step 1: Additive Noise Model**

We model the sparse output as a noisy version of the dense output:

$$\hat{\mathbf{Z}} = \mathbf{Z} + \boldsymbol{\epsilon}, \tag{13}$$

where the noise term

$$\epsilon := f(\mathbf{W} \odot \mathbf{M}, \mathbf{x}) - f(\mathbf{W}, \mathbf{x}) = f(\mathbf{W} \odot (\mathbf{M} - \mathbf{1}\mathbf{1}^{\top}), \mathbf{x}) \tag{14}$$

represents the information loss due to masking. We assume:

- $\epsilon$ is zero-mean, isotropic, and independent of $\mathbf{x}$,

- the noise magnitude is small (i.e., pruning induces small perturbations).

**Step 2: Gaussian Assumption and Joint Distribution**

Assuming the input

$$\mathbf{x} \sim \mathcal{N}(\mathbf{0}, \boldsymbol{\Sigma}_{\mathbf{x}}), \tag{15}$$

both outputs $\mathbf{Z}$ and $\hat{\mathbf{Z}}$ are linear Gaussian transformations of $\mathbf{x}$, thus jointly Gaussian random vectors.

**Step 3: Mutual Information for Jointly Gaussian Variables**

Recall that for jointly Gaussian variables $\mathbf{Z}$ and $\hat{\mathbf{Z}}$, the mutual information is given by

$$I(\mathbf{Z}; \hat{\mathbf{Z}}) = \frac{1}{2} \log \frac{|\boldsymbol{\Sigma}_{\hat{\mathbf{Z}}}|}{|\boldsymbol{\Sigma}_{\hat{\mathbf{Z}}|\mathbf{Z}}|}, \tag{16}$$

where

$$\boldsymbol{\Sigma}_{\hat{\mathbf{Z}}} = \mathrm{Cov}(\hat{\mathbf{Z}}), \quad \boldsymbol{\Sigma}_{\hat{\mathbf{Z}}|\mathbf{Z}} = \mathrm{Cov}(\hat{\mathbf{Z}}|\mathbf{Z}). \tag{17}$$

Since $\hat{\mathbf{Z}} = \mathbf{Z} + \epsilon$ and $\epsilon$ is independent of $\mathbf{Z}$,

$$\boldsymbol{\Sigma}_{\hat{\mathbf{Z}}|\mathbf{Z}} = \boldsymbol{\Sigma}_{\epsilon}, \quad \boldsymbol{\Sigma}_{\hat{\mathbf{Z}}} = \boldsymbol{\Sigma}_{\mathbf{Z}} + \boldsymbol{\Sigma}_{\epsilon}. \tag{18}$$

Therefore,

$$I(\mathbf{Z}; \hat{\mathbf{Z}}) = \frac{1}{2} \log \frac{|\boldsymbol{\Sigma}_{\mathbf{Z}} + \boldsymbol{\Sigma}_{\epsilon}|}{|\boldsymbol{\Sigma}_{\epsilon}|}. \tag{19}$$

**Step 4: Approximation for Small Noise Covariance**

When the noise covariance $\boldsymbol{\Sigma}_{\epsilon}$ is small compared to $\boldsymbol{\Sigma}_{\mathbf{Z}}$, we apply the matrix determinant lemma or first-order Taylor expansion:

$$|\boldsymbol{\Sigma}_{\mathbf{Z}} + \boldsymbol{\Sigma}_{\epsilon}| \approx |\boldsymbol{\Sigma}_{\mathbf{Z}}| \left(1 + \mathrm{Tr}\left(\boldsymbol{\Sigma}_{\mathbf{Z}}^{-1}\boldsymbol{\Sigma}_{\epsilon}\right)\right). \tag{20}$$

Thus,

$$\begin{aligned} I(\mathbf{Z}; \hat{\mathbf{Z}}) &\approx \frac{1}{2} \log \frac{|\boldsymbol{\Sigma}_{\mathbf{Z}}| \left(1 + \mathrm{Tr}(\boldsymbol{\Sigma}_{\mathbf{Z}}^{-1}\boldsymbol{\Sigma}_{\epsilon})\right)}{|\boldsymbol{\Sigma}_{\epsilon}|} \\ &= \frac{1}{2} \left[\log \frac{|\boldsymbol{\Sigma}_{\mathbf{Z}}|}{|\boldsymbol{\Sigma}_{\epsilon}|} + \log \left(1 + \mathrm{Tr}(\boldsymbol{\Sigma}_{\mathbf{Z}}^{-1}\boldsymbol{\Sigma}_{\epsilon})\right)\right]. \end{aligned} \tag{21}$$

Ignoring the weak dependence on the trace term in the small noise regime, the dominant term is

$$I(\mathbf{Z}; \hat{\mathbf{Z}}) \approx \mathrm{const} - \frac{1}{2} \log |\boldsymbol{\Sigma}_{\epsilon}|. \tag{22}$$

**Step 5: Trace Approximation and Connection to MSE**

If $\boldsymbol{\Sigma}_{\epsilon}$ is diagonal or near-diagonal, the log-determinant can be approximated by the trace:

$$\log |\boldsymbol{\Sigma}_{\epsilon}| \approx \mathrm{Tr}(\boldsymbol{\Sigma}_{\epsilon}). \tag{23}$$

Recalling that

$$\mathrm{Tr}(\boldsymbol{\Sigma}_{\epsilon}) = \mathbb{E}_{\mathbf{x}} \left[\|\epsilon\|_2^2\right], \tag{24}$$

we arrive at the key equivalence:

$$I(\mathbf{Z}; \hat{\mathbf{Z}}) \approx \text{const} - \frac{1}{2}\mathbb{E}_{\mathbf{x}}\left\|\mathbf{Z} - \hat{\mathbf{Z}}\right\|_2^2. \tag{25}$$

**Final Conclusion:**

Maximizing the mutual information between dense and sparse outputs is approximately equivalent to minimizing their expected squared difference:

$$\max_{\mathbf{M}\in\mathcal{S}_{n:m}} I\big(f(\mathbf{W}, \mathbf{x}); f(\mathbf{W}\odot\mathbf{M}, \mathbf{x})\big) \quad\Longleftrightarrow\quad \min_{\mathbf{M}\in\mathcal{S}_{n:m}} \mathbb{E}_{\mathbf{x}}\|f(\mathbf{W}, \mathbf{x}) - f(\mathbf{W}\odot\mathbf{M}, \mathbf{x})\|_2^2. \tag{26}$$

This provides an information-theoretic justification for using MSE-based objectives in $n\!:\!m$ sparsity mask learning and pruning.

**Part 2: Stochastic Relaxation via Expected Mask.** To enable differentiable optimization, we introduce a stochastic relaxation. Let the binary mask $\mathbf{M} \in \{0,1\}^d$ be partitioned into $B = d/m$ non-overlapping blocks $\{\mathbf{M}_i\}_{i=1}^B$ of size $m$. Each block is constrained to belong to the $n\!:\!m$ mask set:

$$\mathcal{S}_{n:m} := \{\mathbf{s}_j \in \{0,1\}^m \mid \|\mathbf{s}_j\|_0 = n\}, \quad \text{with } K = |\mathcal{S}| = \binom{m}{n}. \tag{27}$$

Instead of selecting a fixed $\mathbf{M}_i$, we sample it from a categorical distribution over $\mathcal{S}_{n:m}$:

$$\mathbf{M}_i \sim \text{Categorical}(p_i), \quad \text{where } p_i = (p_{i1}, \ldots, p_{iK}) \in \Delta^{K-1}, \quad \sum_{j=1}^K p_{ij} = 1.$$

$$\mathbb{P}[\mathbf{M}_i = \mathbf{s}_j] = p_{ij}, \quad \text{with } \mathbf{s}_j \in \mathcal{S}_{n:m}^{(m)}. \tag{28}$$

The expected value of each block becomes:

$$\mathbb{E}[\mathbf{M}_i] = \sum_{j=1}^K p_{ij}\mathbf{s}_j. \tag{29}$$

Then, the full mask is formed as $\mathbf{M} = [\mathbf{M}_1, \ldots, \mathbf{M}_B] \in \mathcal{S}_{n:m}$, and its expectation is given by:

$$\mathbb{E}_{\mathbf{M}\sim p}[\mathbf{M}] := [\mathbb{E}[\mathbf{M}_1], \ldots, \mathbb{E}[\mathbf{M}_B]] = \Big[\sum_{j=1}^K p_{1j}\mathbf{s}_j, \ldots, \sum_{j=1}^K p_{Bj}\mathbf{s}_j\Big] \in [0, 1]^d \tag{30}$$

Let $\hat{\mathbf{Z}} := f(\mathbf{W}\odot\mathbf{M}, \mathbf{x})$ again denote the sparse output. Taking the expectation over the mask:

$$\mathbb{E}_{\mathbf{M}\sim p}[\hat{\mathbf{Z}}] = \mathbb{E}[f(\mathbf{W}\odot\mathbf{M}, \mathbf{x})] = f(\mathbf{W}\odot\mathbb{E}_{\mathbf{M}\sim p}[\mathbf{M}], \mathbf{x}), \tag{31}$$

using the linearity of expectation and the independence of $\mathbf{x}$ and $\mathbf{M}$. Note that here $f$ denotes a function of MHSA or FFN with linear activation.

The expected squared error between the dense output and the expected sparse output becomes:

$$\mathcal{L}(p) := \mathbb{E}_{\mathbf{x}}\|f(\mathbf{W}, \mathbf{x}) - \mathbb{E}_{\mathbf{M}\sim p}[f(\mathbf{W}\odot\mathbf{M}, \mathbf{x})]\|^2 \tag{32}$$

$$= \mathbb{E}_{\mathbf{x}}\|f(\mathbf{W}, \mathbf{x}) - f(\mathbf{W}\odot\mathbb{E}[\mathbf{M}], \mathbf{x})\|^2 \tag{33}$$

$$= \mathbb{E}_{\mathbf{x}}\left\|f(\mathbf{W}\odot(\mathbf{1}\mathbf{1}^\top - \mathbb{E}[\mathbf{M}]), \mathbf{x})\right\|^2. \tag{34}$$

The relaxation transforms a discrete selection problem into a continuous optimization over categorical parameters $p_i$. When $p_i$ becomes one-hot (i.e., $p_{ij} = 1$ for some $j$), the relaxation recovers a valid deterministic $n\!:\!m$ sparse mask. Thus, optimizing the surrogate loss:

$$\min_{\{p_i\}} \mathbb{E}_{\mathbf{x}}\|f(\mathbf{W}, \mathbf{x}) - f(\mathbf{W}\odot\mathbb{E}[\mathbf{M}], \mathbf{x})\|^2 \tag{35}$$

serves as a smooth surrogate to the original discrete problem:

$$\min_{\mathbf{M} \in \mathcal{S}_{n:m}} \mathbb{E}_{\mathbf{x}} \|f(\mathbf{W}, \mathbf{x}) - f(\mathbf{W} \odot \mathbf{M}, \mathbf{x})\|^2. \tag{36}$$

This surrogate can be optimized using standard gradient-based methods by parameterizing $p_i$ (e.g., via logits and softmax). This result shows that our relaxation bridges discrete sparsity and differentiable training, justifying mask learning as an information-preserving approximation while retaining $n\!:\!m$ structure. This concludes the proof. $\square$

*Remark* 3 (On the Linear and Gaussian Assumptions). Theorem 1 is derived under a linear-layer and Gaussian input assumption to obtain a clean mutual-information-to-MSE relaxation. Although Transformer blocks include nonlinearities (e.g., SwiGLU (Shazeer, 2020)), the justification remains valid in practice for two reasons. First, pruning is performed in a small-perturbation regime, where a first-order Taylor expansion provides an accurate local linearization of the nonlinear layer, allowing the additive-noise and reconstruction-error analysis to apply to the Jacobian-linearized dynamics. Second, the Gaussianity assumption concerns high-dimensional pre-activations (e.g., FFN inputs), which are empirically close to Gaussian due to aggregation through residual connections and the stabilizing effect of LayerNorm (a Central Limit Theorem (CLT)-like phenomenon). Empirically, we observe that the pruning-induced reconstruction error closely follows a Gaussian distribution across layers, further supporting the validity of the approximation in practical LLM settings.

## B  OPTIMIZATION

**Continuous Relaxation via the Gumbel-Softmax Trick.** Optimizing Eq. 12 is challenging because mask variables are sampled from a categorical distribution, making the problem discrete and non-differentiable. To enable end-to-end training, we reparameterize mask selection using the Gumbel-Softmax relaxation (Jang et al., 2017; Maddison et al., 2017). The categorical distribution over $\mathcal{S}$ with logits $\mathbf{p}_{\ell,i}$ is relaxed as:

$$\widehat{\mathbf{M}}_{\ell,i} = \sum_{j=1}^{|\mathcal{S}|} z_{\ell,i,j} \cdot \mathbf{s}_j, \quad z_{\ell,i,j} = \frac{\exp\left((\log p_{\ell,i,j} + \pi_{\ell,i,j})/\tau\right)}{\sum_{j'} \exp\left((\log p_{\ell,i,j'} + \pi_{\ell,i,j'})/\tau\right)}, \tag{37}$$

where $\pi_{\ell,i,j} \sim \text{Gumbel}(0,1)$ and $\tau > 0$ is a temperature controlling the approximation to discreteness. The mask set is $\mathcal{S} = \{\mathbf{s}_1, \ldots, \mathbf{s}_{|\mathcal{S}|}\}$. As $\tau \to 0$, the relaxation converges to discrete selection; during training, $\tau$ is annealed to transition from soft interpolation to near-discrete pruning. With this relaxation, masked weights are updated as $\widehat{\mathbf{W}}_{\ell,i} = \mathbf{W}_{\ell,i} \odot \widehat{\mathbf{M}}_{\ell,i}$. Fig. 2 illustrates the relaxation process, which permits gradient flow through mask selection, enabling gradient-based optimization. Algorithm 1 summarizes the HyperPrune framework.

**Temperature Annealing Schedule in Gumbel-Softmax Relaxation.** In experiments, we use the standard exponential temperature annealing schedule for the Gumbel–Softmax relaxation. The temperature $\tau$ is initialized at a relatively high value to enable smooth gradients and is gradually annealed toward a small, non-zero final value. Concretely, we adopt

$$\tau(t) = \max(\tau_f, \ \tau_0 \cdot \exp(-\eta\, t)),$$

where we set $\tau_0 = 1.0$, $\tau_f = 0.5$, and use a small decay rate $\eta$ (typically in the range $10^{-5}$ to $10^{-4}$), following common practice and the schedule recommended in (Jang et al., 2017). Here $t$ is the current training step (or epoch). This annealing strategy stabilizes early training while progressively encouraging the hypernetwork to produce near-discrete mask selections.

**Optimization Algorithm.** Algorithm 1 details the training procedure of HyperPrune. The algorithm is designed to efficiently generate semi-structured pruning masks through a shared, context-aware hypernetwork while maintaining memory scalability across large LLMs. At each layer, the pretrained weights are loaded onto the GPU in blocks, and the hypernetwork—conditioned on both layer embeddings and task embeddings—produces logits that parameterize a Gumbel-Softmax distribution for sampling structured 2:4 masks. This stochastic relaxation enables differentiable mask learning while respecting hardware-friendly sparsity. During optimization, the model minimizes a

reconstruction loss to preserve activation fidelity, combined with two key regularizations: (i) outlier retention to safeguard critical features, and (ii) continual pruning to enforce cross-layer consistency and knowledge transfer. After training each layer, the hypernetwork parameters are partially frozen to stabilize future pruning decisions, and the corresponding weights are offloaded to reduce memory usage, allowing the method to scale to models as large as LLaMA-2 70B on a single A100 GPU. By jointly leveraging context-aware embeddings, structured mask sampling, and memory-efficient training, HyperPrune directly aligns pruning with both accuracy retention and practical deployability.

## C  TRANSFER LEARNING FOR SPARSE PRIORS

To accelerate HyperPrune training and leverage prior knowledge, we incorporate sparse priors into the relaxed distribution over candidate masks $\mathbf{s}_j \in \mathcal{S}$. Such priors can be obtained from fast heuristic pruning methods like Wanda (Sun et al., 2024). We bias the logits of candidate masks toward a reference prior $\widetilde{\mathbf{M}}$ using cosine similarity:

$$\log p_{\ell,ij} \leftarrow \log p_{\ell,ij} + \gamma \cdot \log \left( \frac{\exp(\beta \cdot \text{cos-sim}(\widetilde{\mathbf{M}}_i, \mathbf{s}_j))}{\sum_{j'} \exp(\beta \cdot \text{cos-sim}(\widetilde{\mathbf{M}}_i, \mathbf{s}_{j'}))} \right), \tag{38}$$

where $\gamma$ controls the prior strength and $\beta$ sharpens the similarity-based softmax. This preserves the probabilistic semantics while softly biasing selection toward masks structurally aligned with the prior. Setting $\gamma = 0$ ignores the prior, whereas larger $\gamma$ interpolates smoothly toward prior-guided pruning.

**Automatically Adjusting $\gamma$.**  To allow the model to gradually transition from relying on the prior to relying on its own learned signal, we anneal the prior influence parameter $\gamma$ over time. Specifically, we decay $\gamma$ according to an exponential schedule:

$$\gamma_t = \gamma_0 \cdot \exp(-\alpha \cdot t), \tag{39}$$

where $\gamma_0$ is the initial prior strength, $\alpha > 0$ is a decay rate hyperparameter, and $t$ is the current training step (or epoch). This annealing scheme ensures that the model initially benefits from the guidance of the prior mask, but progressively shifts toward data-driven learning as training advances.

The decay rate $\alpha$ can be tuned to control how quickly the prior influence vanishes. A smaller $\alpha$ leads to a slower decay, allowing the prior to remain active longer, which is useful in low-data or noisy regimes. This dynamic adjustment supports a smooth interpolation between prior-driven exploration and model-driven convergence.

**Supervised Prior Pretraining.**  In addition to the cosine-similarity logit bias of Eq. 38, the hypernet can be initialized through a supervised pretraining stage (Stage A) in which it is trained to emit the prior mask directly. Concretely, for each (layer, component) the per-row hypernet logits are matched, via cross-entropy, to the index of the Wanda (Sun et al., 2024) or SparseGPT (Frantar & Alistarh, 2023) pattern selected on the same weight group. Stage A runs for 12,000 optimization steps with AdamW (weight decay 0.1) and learning rate `sup_lr` $= 1 \times 10^{-3}$. Each step batches over randomly sampled (layer, component) pairs, drawing 50,000 weight-groups per step; by default all rows of each component are made visible to the hypernet. This supervised initialization is the procedural counterpart of the cosine-similarity bias (Eq. 38) and aligns the hypernet's prior with the chosen reference pruner before the reconstruction objective is engaged. A separate residual one-hot bias $\alpha \cdot \mathbf{1}[\,\text{prior\_idx}\,]$ on the candidate-mask logits before the Gumbel-softmax is applied *only* during the Stage B reconstruction stage (see Appendix D.3); attaching such a bias during Stage A would trivialize the supervised objective, because the cross-entropy target is itself the prior-mask index. The cosine-similarity bias of Eq. 38 and the residual one-hot bias $\alpha$ are not applied simultaneously in any configuration described in this paper.

Table 2: Technical Specifications of the Training Environment (Hardware and Software) used for all Main Paper Experiments.

| GPU | CPU | CUDA | OS | PyTorch |
|-----|-----|------|-----|---------|
| $1\times$ A100 (80GB Memory) | $2 \times$ AMD EPYC 7742 2.25GHz 64 cores | 11.4 | Ubuntu 20.04 | 2.6.0 |

# D  EXPERIMENTS

## D.1  TRAINING ENVIRONMENT

All experiments for HyperPrune are conducted on a single NVIDIA A100 GPU with 80GB memory, supported by dual AMD EPYC 7742 CPUs (64 cores, 2.25GHz). The software stack includes CUDA 11.4, PyTorch 2.6.0, and Ubuntu 20.04, as summarized in Table 2. This environment is representative of widely available high-performance computing clusters used for large-scale model training. The ample GPU memory of the A100 allows efficient handling of models up to 14B parameters without gradient checkpointing or offloading, ensuring stable and reproducible training. Meanwhile, the combination of optimized CUDA kernels and the latest PyTorch backend provides reliable support for semi-structured sparsity and mixed-precision training, both of which are crucial for accelerating pruning and fine-tuning. We note that HyperPrune does not rely on customized kernels beyond the standard PyTorch and CUTLASS implementations, making our method portable to other modern GPU architectures that support 2:4 sparsity.

**Calibration Data.**  Unless otherwise stated, all reconstruction objectives used in Appendix D and onward draw their calibration data from the C4 corpus (Raffel et al., 2020). The default calibration set consists of 128 samples of length 2048 tokens, sampled with a fixed random seed across runs to ensure reproducibility. WikiText-2 is used as the primary evaluation corpus and may, when explicitly noted, also serve as an alternative calibration source. The sample-size sensitivity sweep over $\{32, 64, 128, 256\}$ samples is reported separately in Section 6.

## D.2  COMPARISON METHODS

We evaluate the proposed **HyperPrune** against four heuristic pruning methods: **Magnitude** (Han et al., 2015), **Wanda**[3] (Sun et al., 2024), **SparseGPT**[4] (Frantar & Alistarh, 2023), and **Pruner-Zero**[5] (Dong et al., 2024), as well as one optimization-based pruning baseline: **MaskPro**[6] (Sun et al., 2025). For all methods, we use the publicly available implementations provided by the original authors and follow their recommended configurations. The only exception is **Magnitude**, for which we use the implementation provided in the Wanda library.

## D.3  HYPERPARAMETER CONFIGURATION

Table 3: Training details and hyper-parameters for mask training in HyperPrune.

| Model | Optimizer | #Iterations | Hidden dim | $\lambda_1$ | $\lambda_2$ | Prior | $\alpha$ | $\gamma_0$ |
|-------|-----------|-------------|------------|-------------|-------------|-------|----------|------------|
| LLaMA-2 7B | AdamW(5e-4, wd=0.1) | 2,000 | $\{256, 512\}$ | [1e-3, 1e3] | [1e-3, 1e3] | SparseGPT | 2 | 1e-5 |
| LLaMA-2 13B | AdamW(5e-4, wd=0.1) | 2,000 | $\{256, 512\}$ | [1e-2, 1e2] | [1e-3, 1e3] | SparseGPT | 2 | 1e-5 |
| LLaMA-2 70B | AdamW(5e-4, wd=0.1) | 2,000 | $\{256, 512\}$ | [1e-2, 1e2] | [1e-3, 1e3] | SparseGPT | 2 | 1e-5 |

Table 3 summarizes the hyper-parameters used for training pruning masks in HyperPrune across different model scales. We employ the AdamW optimizer with a learning rate of 5e-4 and weight decay of 0.1, and fix the training schedule to 2,000 steps, which we found sufficient for convergence of mask learning without overfitting to the calibration data. Embeddings in the hypernetwork are

---

[3] https://github.com/locuslab/wanda
[4] https://github.com/IST-DASLab/sparsegpt
[5] https://github.com/pprp/Pruner-Zero
[6] https://github.com/woodenchild95/Maskpro

initialized from a zero-mean Gaussian distribution with variance 0.01 to ensure stable early training. Specifically, the lightweight hypernetwork employs a four-layered feedforward network. We explore two configurations for the hidden layer size: 256 and 512 neurons. The regularization coefficients $\lambda_1$ and $\lambda_2$, which balance continual pruning and feature outlier retention, are selected from broad logarithmic ranges to capture model-specific sensitivities. For LLaMA-2 7B, stronger regularization ($\lambda_1 \in [10^{-3}, 10^3]$) was required to stabilize pruning, whereas for larger models (13B and 70B), a narrower range ($\lambda_1 \in [10^{-2}, 10^2]$) proved more effective, reflecting improved robustness at scale. The prior is initialized with SparseGPT masks to provide a strong starting point, and the scaling factors $\alpha = 2$ and $\gamma_0 = 10^{-5}$ in Eq. 39 are kept fixed across models for consistent sparsity control. It confirms that our method is not overly sensitive to hyperparameter tuning and scales reliably across model sizes.

For architectural conflagration of the hypernetwork in HyperPrune, we performed a small-scale hyperparameter search over the hypernetwork architecture during development. Specifically, we evaluated variants with 1–3 hidden layers and widths $\{128, 256, 512, 1024\}$. This grid search showed that beyond a certain capacity, e.g., more than two hidden layers or widths larger than 512, the performance of HyperPrune remained essentially unchanged, while training cost increased. The 4-layer FFN with 256/512 hidden dimensions therefore represents a stable operating point: it is expressive enough to model structured masks across layers, yet compact enough to maintain efficiency.

Table 4: Additional hyperparameter sweep ranges for HyperPrune, complementary to Table 3. The *Range/Set* column lists the values explored during development; the *Default* column lists the value used in the configurations described elsewhere in the paper. This table reports configuration ranges only and does not reference any reported metric. Entries listed as "[value not provided]" denote sweep dimensions whose production-time default is not pinned down in the rest of the paper.

| Hyperparameter | Range / Set | Default |
| --- | --- | --- |
| Gumbel temperature $\tau$ (cascade) | 0.5 (constant) | 0.5 |
| Stage A supervised steps (sup_steps) | 12,000 (fixed) | 12,000 |
| Stage B inner steps per layer (cascade_inner_steps) | $\{300, 600, 1200\}$ | 300 (7B/13B/L3-8B); 1200 (70B) |
| Samples per inner step (ft_nsamples) | $\{4, 8, 16, 32\}$ | 4 (7B/13B/L3-8B); 16 (70B) |
| Random rows per step (rows_per_step) | $\{200, 400, 800, 1600\}$ | 400 |
| Fixed-row slice size (fixed_rows_count) | $\{100, 200, 300, 400\}$ | 200 |
| Fixed-row slice position (fixed_rows_pos) | $\{$first, last$\}$ | [value not provided] |
| Residual one-hot bias (wanda_residual_alpha) | $\{0.5, 1.0, 2.0, 4.0\}$ | 2.0 |
| 1-D CNN channels (cnn_channels) | $\{64, 96\}$ | [value not provided] |
| 1-D CNN kernel size (cnn_kernel_size) | $\{3, 5, 7\}$ | [value not provided] |
| 2-D CNN channels (cnn2d_channels) | $\{64, 85\}$ | [value not provided] |
| 2-D CNN kernel size (cnn2d_kernel_size) | $\{3, 5\}$ | [value not provided] |
| Prior source (prior_source) | $\{$SparseGPT, Wanda, none$\}$ | SparseGPT |
| Supervised learning rate (sup_lr) | $1 \times 10^{-3}$ (fixed) | $1 \times 10^{-3}$ |
| Fine-tuning learning rate (ft_lr) | $3 \times 10^{-4}$ (fixed) | $3 \times 10^{-4}$ |
| Hessian-diagonal input feature | $\{$enabled, disabled$\}$ | [value not provided] |
| Layer embedding | $\{$enabled, disabled$\}$ | [value not provided] |
| Component embedding | $\{$enabled, disabled$\}$ | [value not provided] |

**Stage-A vs Stage-B Configuration.** The 2,000 iterations reported in Table 3 refer to the *per-layer* iteration budget of the cascade reconstruction stage (Stage B): this budget is consumed independently for each transformer layer. Stage B is preceded by a supervised pretraining stage (Stage A, see Appendix C) which runs for an additional 12,000 steps and is shared across the whole model rather than re-incurred per layer. Both stages use AdamW with weight decay 0.1, but they use different learning rates: sup_lr $= 1 \times 10^{-3}$ for Stage A and ft_lr $= 3 \times 10^{-4}$ for Stage B. The Stage B per-layer inner-step budget cascade_inner_steps is itself a swept hyperparameter drawn from $\{300, 600, 1200\}$, with default 300 for the 7B, 13B, and LLaMA-3 8B configurations and 1200 for the 70B configuration. The number of calibration samples reused inside each inner step, ft_nsamples, is similarly swept over $\{4, 8, 16, 32\}$ with defaults 4 for the 7B/13B/LLaMA-3 8B configurations and 16 for the 70B configuration. The full set of sweep ranges, including those for the alternative hypernetwork architectures and row-sampling regimes, is summarized in Table 4.

### D.4 Context-Aware Embedding Initialization and Optimization

**Initialization.** Jointly optimizing the task embedding $\mathbf{e}_t$ and hypernetwork parameters $\theta$ can be unstable. Careful initialization of $\mathbf{e}_t$ improves convergence and generalization. For initializing $\mathbf{e}_t^{(0)}$, a **one-hot vector with added small Gaussian noise** is recommended ($\mathbf{e}_t^{(0)} = \mathbf{1}_t + \epsilon$, $\epsilon \sim \mathcal{N}(0, \sigma^2\mathbf{I})$, $\sigma \ll 1$). This approach combines the benefits of clear task separation from one-hot encoding with the advantages of smooth optimization provided by the noise, which also helps break symmetry and facilitates early-stage gradient flow. When training, consider applying **norm regularization** ($\lambda\|\mathbf{e}_t\|^2$), using **smaller learning rates** specifically for $\mathbf{e}_t$, and optionally **freezing $\mathbf{e}_t$** during the initial warmup phase.

Unlike discrete and untrainable one-hot embeddings, which create a hard separation between tasks and ignore their relationships, trainable embeddings offer a continuous, dense representation that can be updated with gradients. This allows them to capture the similarity between tasks, leading to improved generalization and scalability. The trainable approach enables soft parameter sharing across tasks, making it a more efficient and effective method for hypernetworks.

**Configuration of Embedding Dimensionality.** In our hyperparameter search, we experimented with several embedding dimensions for both the layer-specific embeddings $\mathbf{e}$ and the component embeddings $\mathbf{t}$, specifically $\{48, 64, 96, 128\}$. We found that once the embedding size exceeds a relatively small threshold (e.g., $d \geq 96$), further increasing the dimensionality brings no noticeable performance improvement while incurring additional computational cost. Based on this study, we recommend using $d = 64$ for the layer embeddings and the component embeddings.

**Optimization of Context-Aware Embeddings.** Both embeddings are learned jointly with the hypernetwork parameters $\theta$ via backpropagation, forming part of the input to $h_\theta$ in Eq. 12. Specifically, for each weight group $\mathbf{W}_{\ell,i}$, the hypernetwork generates a probabilistic $n{:}m$ mask conditioned on the corresponding embeddings:

$$\mathbf{p}_{\ell,i} = \text{Softmax}\big(h_\theta(\mathbf{W}_{\ell,i}, \mathbf{e}_\ell, \mathbf{t})\big),$$

where $\mathbf{t}$ is the component embedding for the current subcomponent. Gradients of the reconstruction loss and regularization terms are backpropagated through $h_\theta$ to update both the hypernetwork parameters and the embeddings, allowing the embeddings to capture layer- and component-specific pruning patterns. This results in a small set of context-aware parameters ($d \times (L + 7)$ in total) that modulate mask generation adaptively across the network.

### D.5 Generalization to LLaMA-3 Models

To assess whether HyperPrune extends beyond LLaMA-2, we further evaluate its performance on the more recent **LLaMA-3 8B** (Grattafiori et al., 2024), which incorporates revised activation functions and architectural updates. As shown in Table 5, HyperPrune maintains strong pruning performance on this modern architecture, achieving a Wikitext-2 perplexity of **15.63**. This surpasses Wanda, SparseGPT, and MaskPro, and closely approaches the Pruner-Zero baseline (15.21). These results indicate that HyperPrune's sparsity patterns and pruning dynamics transfer effectively to contemporary LLM families.

### D.6 Scalability of the Calibration Set

We evaluate the effect of calibration dataset size on zero-shot performance for LLaMA-2-7B, as shown in Table 6. The results indicate a consistent improvement across all tasks as the dataset size increases from 32 to 256 samples. For example, accuracy on BoolQ rises from 64.83% to 70.85%, RTE from 53.95% to 60.46%, HellaSwag from 50.65% to 56.34%, WinoGrande from 59.11% to 64.70%, ARC-Easy from 61.84% to 67.20%, ARC-Challenge from 24.34% to 32.78%, and Open-BookQA from 21.51% to 27.20%. Correspondingly, the overall mean accuracy improves steadily from 48.03% with 32 samples to 54.22% with 256 samples. These findings highlight two key observations. First, while increasing the calibration dataset size consistently boosts performance, the gains are relatively modest, suggesting diminishing returns beyond a few hundred samples. Second, even with only 32 samples, HyperPrune achieves reasonably strong zero-shot performance across

Table 5: Wikitext-2 perplexity of pruning methods applied to LLaMA-3 8B under identical sparsity settings. HyperPrune generalizes well to the updated LLaMA-3 architecture, outperforming prior baselines and closely matching the strong Pruner-Zero method.

| Method | Wikitext-2 PPL |
|---|---|
| LLaMA-3 8B (Dense) | 5.76 |
| Magnitude Pruning | $2.43 \times 10^3$ |
| SparseGPT | 18.38 |
| Wanda | 21.34 |
| Pruner-Zero | **15.21** |
| MaskPro | 16.70 |
| HyperPrune | 15.63 |

Table 6: Zero-shot evaluations of masks trained with different dataset sizes on LLaMA-2-7B.

| Dataset Size | BoolQ ↑ | RTE ↑ | HellaS. ↑ | WinoG. ↑ | ARC-E ↑ | ARC-C ↑ | OBQA ↑ | Mean ↑ |
|---|---|---|---|---|---|---|---|---|
| 32 samples | 64.83 | 53.95 | 50.65 | 59.11 | 61.84 | 24.34 | 21.51 | 48.03 |
| 64 samples | 67.94 | 57.72 | 53.64 | 62.21 | 63.94 | 29.99 | 24.54 | 51.43 |
| 128 samples | 70.69 | 60.22 | 55.28 | 64.50 | 66.32 | 32.55 | 26.76 | 53.76 |
| 256 samples | 70.85 | 60.46 | 56.34 | 64.70 | 67.20 | 32.78 | 27.20 | 54.22 |

diverse tasks, demonstrating its robustness and ability to learn effective masks from limited calibration data. Overall, these results underscore HyperPrune's efficiency in leveraging small datasets while maintaining strong generalization.

## D.7 SUPPLEMENTARY RESULTS FOR ABLATION STUDY

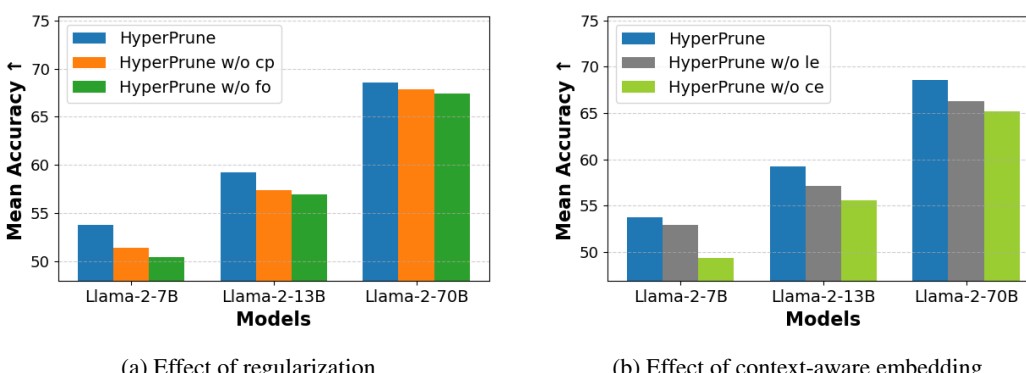

(a) Effect of regularization  (b) Effect of context-aware embedding

Figure 5: Perplexity on Wikitext (a) and Mean Accuracy on seven zero-shot tasks (b) for different pruning methods, evaluated using 1, 2, 4, 8, 16, 32, 64, 128, 256, and 512 calibration samples.

Fig. 5 present ablation studies on HyperPrune on zero-shot tasks, which analyzes the contribution of its core components across LLaMA-2-7B models of different sizes. Panel (a) evaluates the impact of regularization, showing that removing either continual pruning (`w/o cp`) or feature orthogonality (`w/o fo`) generally reduces Mean Accuracy, with the full HyperPrune consistently achieving the best results, especially on the 7B and 70B models. Panel (b) examines the effect of context-aware embeddings, where excluding the learnable embedding (`w/o le`) or context-awareness (`w/o ce`) substantially degrades performance on the 7B and 13B models, confirming their importance for adapting pruning to downstream tasks.

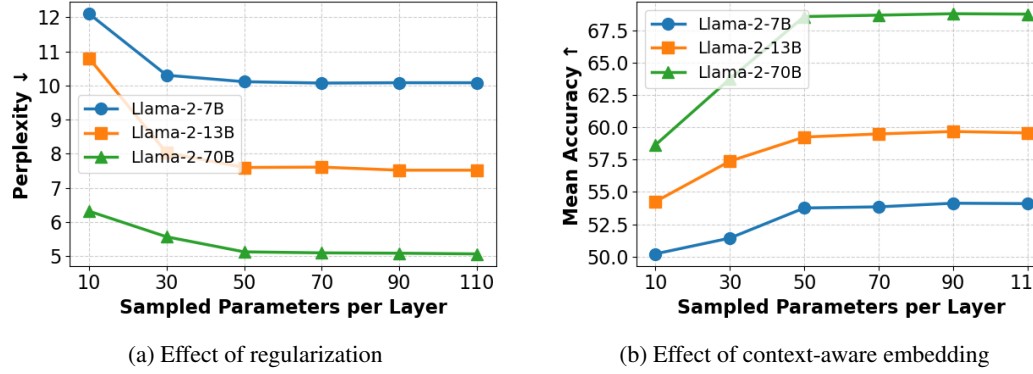

| (a) Effect of regularization | (b) Effect of context-aware embedding |

Figure 6: Perplexity on Wikitext (a) and Mean Accuracy on seven zero-shot tasks (b) for Hyper-Prune, evaluated using 10, 30, 50, 70, 90, and 110 sampled parameters per layer of LLaMA-2-7B.

## D.8 EVALUATION OF THE SAMPLING NUMBER OF PARAMETERS

In Eq. 12, we maintain a set of layer-wise model parameters in the LLM to compute the expectation over $\mathbf{W}$ for continual pruning. Figure 6 explores the effect of pruning density by varying the number of sampled parameters per layer. As expected, perplexity on Wikitext decreases as sparsity is reduced, but plateaus after roughly 50 sampled parameters, while Mean Accuracy on seven zero-shot tasks follows a similar trend. Importantly, LLaMA-2-70B retains strong performance even under high sparsity, highlighting HyperPrune's robustness and efficiency in preserving critical weights across model scales.

## D.9 INFERENCE SPEEDUP SIMULATION.

Table 7: Acceleration of matrix multiplication (in milliseconds) for LLaMA-2-7B with semi-structured 2:4 sparsity obtained by HyperPrune on an NVIDIA A100 GPU.

| LLaMA Layer | Dense (ms) | 2:4 Sparsity (ms) | Speedup |
|---|---|---|---|
| q/k/v/o projection | 2.15 | 1.30 | 1.65× |
| up/gate projection | 5.95 | 3.70 | 1.61× |
| down projection | 6.05 | 3.90 | 1.55× |

We evaluate the practical inference benefits of HyperPrune on an NVIDIA A100 GPU by leveraging hardware-friendly 2:4 structured sparsity. Following (Frantar & Alistarh, 2023), we measure latency for matrix multiplications in linear layers using the high-performance CUTLASS GEMM kernel. As reported in Table 7, the q/k/v/o, up/gate, and down projection layers achieve speedups of 1.55–1.65×, resulting in an average per-layer acceleration of approximately 1.60×. Applied end-to-end on LLaMA-2-7B, this corresponds to a 1.43× reduction in overall inference latency (248 ms → 174 ms). These results demonstrate that HyperPrune's learned pruning masks not only induce structured sparsity but also provide substantial computational efficiency on modern accelerators, highlighting their practical value for real-world deployment.

## D.10 MEMORY SCALABILITY

Table 8: Memory (GB) required for training with 2:4 sparsity.

| Model | MaskLLM | MaskPro | HyperPrune (est.) |
|---|---|---|---|
| LLaMA-2 7B | 339.56 | 39.36 | 15.46 |
| LLaMA-2 13B | 630.80 | 68.41 | 21.63 |
| LLaMA-2 70B | – | – | 33.74 |

Table 8 reports the memory required for training LLaMA-2 models (7B, 13B, 70B) under 2:4 sparsity across 8×A100 GPUs. The results highlight a sharp contrast in scalability among methods. **MaskLLM** suffers from extremely high memory overhead due to its dense mask parameterization, exceeding 600 GB for LLaMA-2 13B and failing with out-of-memory (OOM) errors at 70B scale, making it impractical for very large models. **MaskPro**, in contrast, maintains nearly constant memory consumption because of its lightweight linear probability modeling, requiring only 39–68 GB for 7B–13B. However, it cannot scale to the 70B model within a single-node setting. **HyperPrune** achieves a better balance: by using a compact shared hypernetwork with context-aware embeddings and layer-wise memory offloading, it incurs only modest overhead compared to MaskPro, requiring 15.46 GB for 7B and 21.63 GB for 13B, while remaining feasible even at 70B with 33.74 GB. These results demonstrate that HyperPrune is both memory-efficient and scalable, enabling structured pruning of trillion-scale models on realistic hardware budgets, unlike methods such as MaskLLM that become prohibitive at large scales.

### D.11 GPU-HOUR AND WALL-CLOCK COSTS FOR MASK TRAINING.

Table 9 provides a detailed comparison of computational and memory overhead for $2:4$ sparsity on LLaMA-2 models, reporting GPU hours and memory usage for mask training under identical calibration sizes. As shown, HyperPrune has significantly lower computational and memory costs compared to MaskPro and MaskLLM. For LLaMA-2 7B, MaskLLM requires $\approx 1200$ GPU hours and 339.56 GB of memory across 8× A100 GPUs, while MaskPro uses 10 GPU hours and 39.36 GB. HyperPrune achieves the same task using only 7 GPU hours and 15.46 GB. Similarly, for LLaMA-2 13B, MaskLLM takes $\approx 2300$ GPU hours, MaskPro 22 GPU hours, whereas HyperPrune only 15 GPU hours with 21.63 GB. We do not report LLaMA-2 70B here because MaskLLM and MaskPro cannot complete pruning on a single A100 GPU. These results demonstrate HyperPrune's practical advantage in both computation and memory.

Table 9: GPU hours and memory usage required for training pruning masks under $2:4$ structured sparsity on LLaMA-2 models. HyperPrune achieves the lowest computational and memory cost across all settings, while MaskLLM requires orders of magnitude more resources and fails to scale to larger models.

| Method | LLaMA-2 7B | | LLaMA-2 13B | |
|---|---|---|---|---|
| | Memory (GB) | GPU Hours | Memory (GB) | GPU Hours |
| MaskLLM | 339.56 | $\approx 1200$ | 630.80 | $\approx 2300$ |
| MaskPro | 39.36 | 10 | 68.41 | 22 |
| **HyperPrune** | **15.46** | **7** | **21.63** | **15** |

### D.12 ABLATION STUDY: EFFECT OF PRIOR-BASED INITIALIZATION

We investigate the impact of prior-based initialization on HyperPrune by testing three variants: initialization from a **SparseGPT prior** (Frantar & Alistarh, 2023), from a **Wanda prior** (Sun et al., 2024), and with **no prior**. Experiments were conducted on LLaMA-2-7B and LLaMA-2-13B using Wikitext-2 perplexity. As shown in Table 10, prior-based initialization consistently improves pruning quality, with SparseGPT providing the strongest gains. These results indicate that HyperPrune effectively leverages upstream sparsity patterns while remaining robust to the choice of prior.

Table 10: Ablation on prior-based initialization for HyperPrune. Using a SparseGPT or Wanda prior improves Wikitext-2 perplexity on both LLaMA-2-7B and 13B, while HyperPrune without a prior performs worse, indicating the benefit—but not strict necessity—of prior information.

| Initialization Method | LLaMA-2 7B | LLaMA-2 13B |
|---|---|---|
| HyperPrune + SparseGPT prior | **10.11** | **7.60** |
| HyperPrune + Wanda prior | 10.28 | 7.79 |
| HyperPrune (no prior) | 10.75 | 8.04 |

### D.13 HYPERNETWORK ARCHITECTURE FAMILY

Beyond the 4-layer feed-forward backbone documented in Appendix D, several alternative hyper-network backbones were considered during development. All variants share the same input/output convention: they ingest a per-row feature vector and emit logits over the candidate 2:4 mask patterns (or a single scalar, in the scalar variant below). They are presented here as architectural documentation only; no comparison table or relative ranking is reported.

**Feed-forward (FFN) backbone.** The default backbone is a 4-layer fully connected network. Its hidden dimension is explored over $\{128, 256, 512, 1024\}$, with the production configurations restricted to $\{256, 512\}$ as listed in Table 3.

**1-D convolutional backbone (`cnn`).** A 1-D convolutional network that treats the per-row feature vector as a length-$d$ sequence. The number of channels is swept over $\{64, 96\}$ and the kernel size over $\{3, 5, 7\}$.

**2-D convolutional backbone (`cnn2d`).** A 2-D convolutional network that operates on a reshaped feature map. The number of channels is swept over $\{64, 85\}$ and the kernel size over $\{3, 5\}$.

**Scalar variant.** As an ablation, a single-scalar (1-d output) lightweight hypernet is also considered. It emits a single logit rather than a categorical distribution over candidate patterns, and is intended to probe the minimum capacity that the hypernet must expose for the reconstruction objective to remain useful.

**Optional input features.** An optional Hessian-diagonal feature can be concatenated to the hypernet input: when enabled, the diagonal entries of the SparseGPT (Frantar & Alistarh, 2023) second-order estimate computed on the current weight group are appended to the per-row feature vector. Independently, a layer-index embedding and a component-type embedding may each be switched on or off; when enabled, both are looked up from learned tables and added to the hypernet input. The four switches (architecture family, Hessian-diagonal feature, layer embedding, component embedding) are configurable independently of one another.

### D.14 ROW-SAMPLING STRATEGY AT STAGE B

The cascade reconstruction stage (Stage B) supports two row-sampling regimes governing how the hypernet's output is exposed to the per-layer MSE objective.

**Random mini-batch regime.** At each inner step, a uniformly random subset of output rows of size `rows_per_step` $\in \{200, 400, 800, 1600\}$ is sampled, and the hypernet emits per-row mask logits only for those rows; the remaining rows are held at their current mask. The default is `rows_per_step` $= 400$. Over many inner steps every row is visited multiple times, so the hypernet ultimately modulates the full output dimension of the component.

**Fixed-row regime.** Alternatively, the hypernet modulates only a deterministic contiguous slice of output rows of size `fixed_rows_count` $\in \{100, 200, 300, 400\}$ (default 200). The slice position `fixed_rows_pos` may be either the first `fixed_rows_count` rows or the last `fixed_rows_count` rows of the component. Rows outside the slice are held at the prior mask (Wanda or SparseGPT) throughout Stage B and are not optimized by the hypernet. This regime is useful for isolating how locally the hypernet contributes mask refinement relative to the underlying prior.

### D.15 SKIP-BOUNDARY-LAYER CONFIGURATION

As an optional configuration, HyperPrune supports keeping a contiguous block of front layers and/or last layers of the transformer fully *dense* while applying strict 2:4 sparsity to the remaining (interior) layers. We use the notation "`fXlY`" to denote a configuration in which the first $X$ and the last $Y$ transformer layers are left dense. Hence `f2l0` keeps the first two layers dense and prunes all others to 2:4, `f0l2` keeps only the last two dense, `f2l2` keeps both ends dense, and `f0l0` recovers the strict 2:4 default reported in the main text.

Under this configuration the effective weight-level sparsity is *strictly below* $0.50$ by an amount determined by the dense-layer count and the total transformer depth: if $L$ denotes the total number of transformer layers and $D = X + Y$ the number of dense boundary layers, the effective sparsity is approximately $0.5\,(L - D)/L$ (ignoring per-layer parameter-count differences across attention and MLP blocks). This setting is provided as an optional knob for deployments that can tolerate slightly less than $0.50$ sparsity in exchange for protecting the boundary layers; it is not the default configuration reported in the main text, where every transformer layer is pruned to strict 2:4. In practice, we recommend skipping the first two layers of the LLM (i.e., the `f2l0` setting), as we have empirically observed that the earliest transformer layers tend to be the most sensitive to pruning and leaving them dense recovers most of the quality gap at a very small sparsity cost.

### D.16 LAYER-WISE WEIGHT COMPENSATION

HyperPrune optionally supports layer-wise weight compensation analogous to the error-compensation procedures used in GPTQ (Frantar et al., 2022a) and SparseGPT (Frantar & Alistarh, 2023). Three independent flags control the behaviour:

- `compensated_propagation`: after a transformer layer has been pruned, the residual reconstruction error (the difference between the original dense layer's output and the output of the pruned layer on the calibration activations) is propagated into the next layer's input activations. Subsequent layers are then optimized against these compensated inputs rather than against the original calibration activations.

- `use_weight_compensation`: an in-place compensation step is applied to the remaining unpruned weights inside the just-pruned layer, in the spirit of the SparseGPT closed-form update, to reduce the immediate per-layer reconstruction error before the hypernet moves on to the next layer.

- `train_on_compensated`: during Stage B, the hypernet itself is trained using the compensated inputs produced by `compensated_propagation`, rather than using the raw calibration activations of the original dense model.

These three flags can be toggled independently or jointly; in practice they are typically enabled together so that activations, weights, and the hypernet objective are mutually consistent across the cascade.

