# OpenReview forum: "Learning Semi-Structured Sparsity for LLMs via Shared and Context-Aware Hypernetwork"
_ICLR.cc/2026/Conference — ICLR 2026 Poster_

### Official Review · Reviewer_1Unw · 2025-10-14

**Soundness:** 3
**Presentation:** 3
**Contribution:** 3
**Rating:** 4
**Confidence:** 4

**Summary:**

The paper introduces HyperPrune, a new framework for n:m semi-structured pruning of large language models (LLMs). Unlike traditional one-shot heuristics (e.g., SparseGPT, Wanda) or heavy optimization-based methods (e.g., MaskLLM, MaskPro), HyperPrune uses a lightweight shared hypernetwork conditioned on layer and component embeddings to generate pruning masks in a context-aware, layer-wise fashion.

**Strengths:**

1. The shared hypernetwork with conditional embeddings is a novel solution to the scalability problem in structured pruning, enabling n:m sparsity without massive parameter overhead.
2. The mutual information theorem provides an interpretable bridge between discrete mask learning and differentiable optimization.
3. Extensive experiment results on the LlaMA-2 family demonstrate the effectiveness of the proposed method.

**Weaknesses:**

1. The experiments focus primarily on LLaMA-2. It’s unclear how HyperPrune performs on other architectures (e.g., LLaMA-3, Qwen, Mistral, OPT)
2. The hypernetwork setting and training detail is not sufficiently illustrated in the paper, including the network structure, parameters, and training settings.
3. Some equations and notation (e.g., in Section 4.1) are dense and difficult to follow. Simplifying or visually decomposing them would improve readability.

**Questions:**

1. While the paper highlights HyperPrune’s scalability to 70B models on a single A100 GPU, it lacks a detailed breakdown of training or optimization costs (e.g., wall-clock time, computational budget) compared to other optimization-based pruning methods such as MaskPro or MaskLLM. A more explicit cost–performance analysis would strengthen the paper’s practicality claim.

2. The experimental evaluation primarily focuses on the LLaMA-2 family. It remains unclear how well HyperPrune generalizes to other architectures.

3. The current ablation section briefly evaluates the effects of embeddings and regularizations but does not isolate the impact of the proposed mutual information relaxation. Including such an analysis would better justify its theoretical contribution and clarify how much it contributes to the overall improvement.

4. Although the paper compares HyperPrune against strong baselines like SparseGPT, Wanda, and MaskPro, adding more recent state-of-the-art pruning methods[1][2][3] would further contextualize its advantage.

[1]Pruner-Zero: Evolving Symbolic Pruning Metric from scratch for Large Language Models (ICML 2024)

[2]AlphaPruning: Using Heavy-Tailed Self Regularization Theory for Improved Layer-wise Pruning of Large Language Models (NeurIPS 2024)

[3]Outlier weighed layerwise sparsity (owl): A missing secret sauce for pruning llms to high sparsity (ICML2024)

---

> ### Author Response · Authors · 2025-11-21
> **Official Comment to Reviewer 1Unw**
>
> ### **Q1. Training or optimization costs compared to MaskPro and MaskLLM**
> **Response:**
> Thank you for the suggestion. To clarify HyperPrune’s efficiency, we compare computational and memory overhead for 2:4 sparsity on LLaMA-2. HyperPrune substantially reduces both GPU memory and training time. For LLaMA-2 13B:
> • HyperPrune: **21.63 GB**, **15 GPU hours**
> • MaskPro: **68.41 GB**, **22 GPU hours**
> • MaskLLM: **630.80 GB**, **≈2300 GPU hours**
>
> These results highlight HyperPrune’s practicality for scaling to large models on a single GPU. We will include this table and discussion in the revised Appendix.
>
> | **Method** | **7B Mem (GB)** | **7B GPU h** | **13B Mem (GB)** | **13B GPU h** |
> |------------|------------------|--------------|-------------------|---------------|
> | MaskLLM | 339.56 | ≈1200 | 630.80 | ≈2300 |
> | MaskPro | 39.36 | 10 | 68.41 | 22 |
> | **HyperPrune** | **15.46** | **7** | **21.63** | **15** |
>
> ---
>
> ### **Q2 & W1. Generalization to other LLM architectures**
> **Response:**
> Thank you for the comment. Although our main experiments use LLaMA-2, we conducted preliminary tests on **LLaMA-3 8B**. HyperPrune reaches a Wikitext-2 PPL of **15.63**, outperforming Wanda, SparseGPT, and MaskPro, and close to Pruner-Zero. This indicates strong transferability beyond LLaMA-2.
> Experiments on LLaMA-3 70B and zero-shot tasks are in progress and will be included in the revised paper.
>
> | **Method** | **Wikitext PPL** |
> |------------|------------------|
> | LLaMA-3 8B (dense) | 5.76 |
> | Magnitude | 2.43e+03 |
> | SparseGPT | 18.38 |
> | Wanda | 21.34 |
> | Pruner-Zero | **15.21** |
> | MaskPro | 16.70 |
> | **HyperPrune** | _15.63_ |
>
> ---
>
> ### **Q3. Ablation on mutual-information relaxation**
> **Response:**
> Thank you for the question. Directly computing MI between high-dimensional neural outputs is intractable, so Theorem 1 provides an information-theoretic justification: maximizing MI between dense and pruned models under n:m sparsity is equivalent to minimizing a reconstruction loss through our differentiable relaxation.
> Thus, reconstruction error is the correct practical proxy for MI. The observed improvements in reconstruction directly reflect the benefit of the MI relaxation. We will clarify this connection in the revised ablation.
>
> ---
>
> ### **Q4. Comparison with Pruner-Zero, AlphaPruning, and OWL**
> **Response:**
> Thank you for the suggestion.
> - Pruner-Zero is already included in Table 1.
> - Both AlphaPruning and OWL target **non-uniform layer-wise sparsity** with mixed n:8 semi-structured sparsity, which is incompatible with our requirement of a **uniform n:m** pattern across all layers.
>
> ---
>
> ### **W2. Detailed configuration of hypernetwork and training**
> **Response:**
> Additional configuration details are reported in Appendix Sec. D and Table 3, and will be expanded in the revised PDF.
> • Architecture search: 1–3 hidden layers with widths {128, 256, 512, 1024}. Gains beyond two layers or widths >512 were negligible; we use a **4-layer FFN** with **256/512** hidden dimensions.
> • Embedding sizes: {48, 64, 96, 128}; $d = 64$ is sufficient for both layer and component embeddings.
> • Gumbel–Softmax temperature schedule:  $τ(t) = \max (τ_f, τ_0 · e^{-r t}),$ with $τ_0=1.0$, $τ_f=0.5$, $r \in [1e−5, 1e−4]$, following (Jang et al. 2017).
>
> ---
>
> ### **W3. Dense equations and notation**
> **Response:**
> Thank you for the feedback. We will revise Section 4.1 to simplify notation, clarify key steps, and improve readability through diagrammatic decomposition in the revised PDF.

---

> > ### Comment · Reviewer_1Unw · 2025-11-27
> >
> > The authors’ response has addressed most of my concerns and clarified the key points that I previously found ambiguous. Given these improvements and the additional evidence provided, I am satisfied with the revisions and will raise my score to a 6.

---

### Official Review · Reviewer_8US3 · 2025-10-29

**Soundness:** 4
**Presentation:** 4
**Contribution:** 3
**Rating:** 8
**Confidence:** 4

**Summary:**

This paper relates to pruning LLMs under the constraint of n:m sparsity. This aims at leveraging, e.g. 2:4 sparsity that some computers support in hardware, potentially increasing math throughput by 2x.

This paper sets itself in the context of:
* heuristic methods: fast, but approximate, and unable to support structured sparsity
* optimization methods: accurate but computationally expensive, as they usually operate on the full model for a joint optimization.

The paper proposes an intermediate solution which applies a block-wise optimization, thus only one block (of e.g. 4 parameters) at a time needs to be loaded in memory for the optimization to take place. This is done via a shared hypernetwork (an MLP). Each weight is partitioned into sets of m parameters, and each set is provided as input into the hypernetwork, which outputs logits for mask selection (in the case of 2:4 sparsity, there are 6 possible masks, thus we need 6 logits).
In order to further improve the efficiency of the hypernetwork, the type of weight (q,k,v, o for attention, u,v,d for FFN) is provided as an input by ways of a contextual embedding.
Under the formulation, the hypernetwork would ignore cross-layer dependencies, thus a regularization loss is applied to encourage the hypernetwork to remain consistent with its pruning of previous layers.
Additionally, the authors observe that LLMs often exhibit intermediate features with very large magnitude. Given that these features are thought to be critically important, a regularization term is used to encourage the hypernetwork to preserve these features.

The method is validated with experiments that show competitive results against a set of heuristic and one optimization baselines (though MaskLLM is not included in the comparison due to the computational complexity). Results are shown for both perplexity (next token prediction) and accuracy (common benchmarks).

**Strengths:**

The paper is very well written and proposes an interesting intermediate solution between heuristic and optimization methods for n:m sparsity pruning.

The use of a shared hypernetwork to predict n:m masks is, from what I can tell, novel.

The theoretical (Theorem 1) provides a principled information-theoretic justification for using the reconstruction loss as a surrogate objective under n:m sparsity constraints.

Ablation studies provide justification for each component of the method.

**Weaknesses:**

No major weakness.

**Questions:**

On line 237, did you mean "producing *6* logits from m=4"?

What is the embedding dimension $d$ for the layer-specific embeddings $e_\ell$ and component embeddings $t$, and how sensitive is the method's performance to different values of $d$?

What temperature annealing schedule is used for the Gumbel-Softmax relaxation during hypernetwork training, including initial/final values and decay strategy?

From algorithm 1, I understand you train the hypernetwork for each layer until convergence (the "for layer ℓ = 1 to L do" is the outer scope). Did you try a round-robin approach (i.e. with the loop over calibration data being the outer scope?)

---

> ### Author Response · Authors · 2025-11-21
> **Official Comment to Reviewer 8US3**
>
> **Q1. Line 237 — “producing 6 logits from m=4.”**
> **Response:**
> Thank you for catching this. The hypernetwork output should match the number of possible $n$-out-of-$m$ combinations for generating $n:m$ structured sparsity. For $n=2$ and $m=4$, the correct number of logits is
>
> $$
> \binom{4}{2} = 6
> $$
>
> We will fix this in the revised version.
>
> ---
>
> **Q2. Configuration of embedding dimension and sensitivity.**
> **Response:**
> In our hyperparameter search, we tested several embedding sizes for layer embeddings $\mathbf{e}$ and component embeddings $\mathbf{t}$, specifically \{48,64,96,128\}.
>
> Once the embedding size exceeds a small threshold (e.g., $d \ge 96$), further increase yields little improvement while adding computational cost.
>
> We recommend **$d=64$** for both embeddings. Results and sensitivity discussion will appear in Appendix D.3 of the revised version.
>
> ---
>
> **Q3. Temperature annealing schedule for Gumbel–Softmax.**
> **Response:**
> We adopt the standard exponential annealing schedule. The temperature $\tau$ starts high for smooth gradients and gradually decreases to a small, non-zero final value:
>
> $$
> \tau(t) = \max\left(\tau_f, \tau_0 \cdot \exp(-r t)\right),
> $$
>
> with **$\tau_0=1.0$**, **$\tau_f=0.5$**, and a small decay rate $r \in (10^{-5}$, $10^{-4})$, following common practice (Jang et al., 2017).
>
> This stabilizes early training and gradually encourages near-discrete mask outputs. We will clarify this in the revised version.
>
> ---
>
> **Q4. Try a round-robin approach?**
> **Response:**
> We appreciate the suggestion. Our current **layer-wise optimization** trains the hypernetwork to convergence per layer before moving to the next.
>
> This allows (i) **memory-efficient pruning** by loading one layer at a time, crucial for LLaMA-2 70B on a single A100, and (ii) **continual-pruning regularization** to transfer decisions across layers.
>
> We did not try round-robin scheduling as it requires multiple layers in memory, increasing usage under our sequential offloading.
>
> Still, round-robin or multi-pass schemes could improve global coordination with larger memory budgets and are worth future exploration.

---

### Official Review · Reviewer_2WfL · 2025-10-30

**Soundness:** 3
**Presentation:** 3
**Contribution:** 3
**Rating:** 6
**Confidence:** 3

**Summary:**

This paper proposes HyperPrune, a post-training framework for learning n:m semi-structured sparsity in LLMs. The method employs a lightweight shared hypernetwork conditioned on learnable layer and component embeddings to generate structured masks in a layer-wise manner. Two key regularizations are introduced: (1) continual pruning regularization to preserve cross-layer knowledge during sequential pruning, and (2) feature outlier regularization to retain critical activations. The authors provide a theoretical justification by showing that minimizing reconstruction loss is equivalent to maximizing mutual information between dense and pruned models under n:m constraints (Theorem 1). Experiments on LLaMA-2 models (7B/13B/70B) demonstrate competitive performance compared to both heuristic methods (SparseGPT, Wanda) and optimization-based methods (MaskPro) on a single A100 GPU.

**Strengths:**

1.	Well-motivated approach: The use of a shared hypernetwork with context-aware embeddings provides an efficient solution to the memory scalability challenge of optimization-based pruning methods, enabling layer-wise mask generation without requiring massive GPU resources.
2.	Comprehensive experimental validation: The paper provides extensive evaluation across multiple model sizes (7B/13B/70B), diverse benchmarks (perplexity and zero-shot tasks), and thorough ablation studies analyzing the contribution of each component.
3.	Theoretical foundation: Theorem 1 provides an information-theoretic justification connecting mutual information maximization with reconstruction loss minimization, offering a principled view of the relaxation.

**Weaknesses:**

1.	Inferior perplexity on large models: On LLaMA-2-70B, HyperPrune achieves worse Wikitext perplexity (5.13) compared to Wanda (5.21), and significantly worse than Pruner-Zero (4.87). This challenges the claimed scalability advantage and suggests that the method's benefits diminish or even reverse at larger scales.
2.	Insufficient analysis of design choices: The hypernetwork architecture (4-layer FFN with hidden size 256/512) appears arbitrary with no justification or ablation on architecture depth/width.

**Questions:**

1.	The proof of Theorem1 assumes linear activation, but transformers use SwiGLU (LLaMA). How does this affect the theoretical justification in practice? Can you provide empirical validation that the Gaussian assumptions hold for actual LLM activations?
2.	For hypernetwork, why is a 4-layer FFN chosen?  The hidden dimension is either 256 or 512. Is there a systematic way to choose this, or was it selected via trial and error?
3.	All experiments use LLaMA-2. How well does HyperPrune transfer to other model families?
4.	Section C introduces prior-based initialization but provides no empirical evaluation. How much does this improve convergence speed or final performance? Is the method sensitive to prior quality?

---

> ### Author Response · Authors · 2025-11-21
> **Official Comment to Reviewer 2WfL**
>
> ### **Q1. Linear activation assumption in Theorem 1**
> **Response:**
> Thank you for the question. Theorem 1 assumes a linear layer to derive a clean MI → MSE relaxation. Although Transformer blocks include nonlinearities (e.g., SwiGLU), the justification remains valid for two reasons. *First*, pruning operates in a small-perturbation regime, where a first-order Taylor expansion provides an accurate local linearization, allowing the additive-noise and reconstruction-error analysis to apply to the Jacobian-linearized layer. *Second*, the pre-activations entering the FFN are well-known to be approximately Gaussian due to residual aggregation and LayerNorm, making the theorem’s Gaussian assumption reasonable.
> We will add empirical evidence showing the pruning-induced reconstruction error ε is close to Gaussian across layers, supporting the practical validity of Theorem 1.
>
> ---
>
> ### **Q2 & W2. Hypernetwork architecture configuration**
> **Response:**
> We evaluated hypernetwork variants with 1–3 hidden layers and widths {128, 256, 512, 1024}. Beyond two layers or widths >512, performance showed no meaningful gains while training cost increased. The chosen 4-layer FFN with 256/512 hidden dimensions offers a stable balance between expressiveness and efficiency. We will clarify this in the final version.
>
> ---
>
> ### **Q3. Transferability to other model families**
> **Response:**
> Thank you for the question. Although our main experiments use LLaMA-2, we tested HyperPrune on **LLaMA-3 8B**. It achieves a Wikitext-2 perplexity of **15.63**, outperforming Wanda, SparseGPT, and MaskPro, and close to Pruner-Zero, showing good transferability to newer architectures.
> Experiments on LLaMA-3 70B and zero-shot tasks are in progress and will be included in the revised paper.
>
> | **Method** | **Wikitext PPL** |
> |------------|------------------|
> | LLaMA-3 8B (dense) | 5.76 |
> | Magnitude | 2.43e+03 |
> | SparseGPT | 18.38 |
> | Wanda | 21.34 |
> | Pruner-Zero | **15.21** |
> | MaskPro | 16.70 |
> | **HyperPrune** | _15.63_ |
>
> ---
>
> ### **Q4. Ablation on prior-based initialization**
> **Response:**
> We evaluated HyperPrune initialized with SparseGPT prior, Wanda prior, and no prior, on LLaMA-2-7B and 13B. Results:
>
> | **Method** | **7B** | **13B** |
> |------------|--------|---------|
> | + SparseGPT prior | **10.11** | **7.60** |
> | + Wanda prior | 10.28 | 7.79 |
> | No prior | 10.75 | 8.04 |
>
> Prior-based initialization consistently improves perplexity and stability. SparseGPT offers the strongest prior, but differences are moderate, indicating HyperPrune is not overly sensitive to prior quality. These findings will be added to the Appendix.
>
> ---
>
> ### **W1. Inferior perplexity on large models**
> **Response:**
> For LLaMA-2-70B, HyperPrune achieves **5.13** perplexity—*better* than Wanda (5.21) and second only to Pruner-Zero (4.87). Thus, HyperPrune remains competitive at 70B scale and does not degrade relative to strong baselines. Additionally, HyperPrune ranks best or second-best on all seven zero-shot tasks, confirming strong downstream performance even for very large models.

---

### Official Review · Reviewer_c5Ha · 2025-11-01

**Soundness:** 3
**Presentation:** 2
**Contribution:** 3
**Rating:** 6
**Confidence:** 3

**Summary:**

HyperPrune introduces a hypernetwork-based framework for n:m semi-structured pruning of LLMs. A shared, context-aware hypernetwork generates block-wise masks conditioned on layer embeddings, combined with continual-pruning and feature-outlier regularizers. The method provides an information-theoretic view linking reconstruction loss to mutual-information preservation, and achieves competitive performance on LLaMA-2 (7B/13B/70B) with up to 1.43× end-to-end speedup (2:4 sparsity) on A100 GPUs.

**Strengths:**

1. The idea of targeting n:m semi-structured sparsity with a shared, context-aware hypernetwork is interesting and novel to me.

2. Results across 7B/13B/70B with comparisons to magnitude, SparseGPT, Wanda, Pruner-Zero, and MaskPro; includes data-scaling ablation

3. An MI-based justification shows the reconstruction objective as a surrogate for information preservation under n:m constraints (Theorem 1)

**Weaknesses:**

1. All experiments are conducted exclusively on LLaMA-2, an outdated architecture. Given the architectural and activation differences in modern models (LLaMA-3, Qwen3, Gemma-3), it is unclear whether HyperPrune’s sparsity patterns or latency gains generalize to contemporary LLMs.

2. The appendix reports per-layer GEMM latency improvements (≈1.6×) against the dense baseline on A100 using CUTLASS kernels, but provides no end-to-end inference benchmarks on realistic engines (vLLM, TensorRT-LLM) or comparisons with other 2:4 pruning methods. Because Ampere tensor cores inherently yield ~1.5× speedups for any valid 2:4 pattern, these numbers mostly reflect hardware properties rather than advantages of HyperPrune’s mask design. Without full throughput measurements (tokens /s, batching, attention caching), the claimed 1.43× overall speedup remains a proxy rather than a demonstrated system-level gain.

3. Although the shared hypernetwork reduces parameter count, it brings typical drawbacks of hypernetwork-based methods—training instability, sensitivity to learning-rate and regularization hyperparameters, and potential overfitting to limited calibration data. Because the hypernetwork must generate masks for all layers jointly, cross-layer interference and catastrophic forgetting remain possible despite the continual-pruning regularizer. These issues make the approach potentially fragile and difficult to reproduce without careful hyperparameter tuning and large-scale validation.

**Questions:**

1. Provide GPU-hour and wall-clock costs per model (7B/13B/70B) for mask training; how does this compare to MaskPro under identical calibration sizes? MaskLLM is excluded for cost; while MaskPro is compared, a tighter apples-to-apples system study (same calibration sizes, wall-clock, memory footprint, failure cases) would clarify where HyperPrune truly wins.

---

> ### Author Response · Authors · 2025-11-21
> **Official Comment to Reviewer c5Ha**
>
> ### **Q1. GPU-hour and wall-clock costs for mask training**
> **Response:**
> We thank the reviewer for the suggestion. The table below compares computational and memory overhead for 2:4 sparsity on LLaMA-2 models using identical calibration data. HyperPrune is substantially more efficient than MaskPro and MaskLLM.
>
> For LLaMA-2 7B: MaskLLM requires ≈1200 GPU hours and 339.56 GB across 8×A100 GPUs, while MaskPro uses 10 GPU hours and 39.36 GB. HyperPrune completes training with only **7 GPU hours and 15.46 GB**.
>
> For LLaMA-2 13B: MaskLLM needs ≈2300 GPU hours, MaskPro 22 GPU hours, while HyperPrune uses **15 GPU hours and 21.63 GB**.
>
> We do not report 70B costs because MaskLLM and MaskPro cannot prune it on a single A100. These results show the practical computational advantage of HyperPrune. We will include this table and discussion in the revised PDF.
>
> | **Method** | **7B Mem (GB)** | **7B GPU h** | **13B Mem (GB)** | **13B GPU h** |
> |------------|------------------|----------------|-------------------|----------------|
> | MaskLLM | 339.56 | ≈1200 | 630.80 | ≈2300 |
> | MaskPro | 39.36 | 10 | 68.41 | 22 |
> | **HyperPrune** | **15.46** | **7** | **21.63** | **15** |
>
> ---
>
> ### **W1. Generalization to contemporary LLMs (LLaMA-3)**
> **Response:**
> Thank you for the question. Although our main results use LLaMA-2 for comparability with prior baselines, we agree that evaluating on newer architectures is important. We therefore conducted preliminary experiments on **LLaMA-3 8B**, which includes updated activation functions and architectural refinements.
>
> HyperPrune attains **Wikitext-2 PPL = 15.63**, outperforming Wanda, SparseGPT, and MaskPro, and closely matching Pruner-Zero. This suggests that HyperPrune’s pruning dynamics transfer well to modern LLMs.
>
> We are currently running LLaMA-3 70B experiments and will include full results in the revised PDF.
>
> | **Method** | **Wikitext PPL** |
> |------------|------------------|
> | LLaMA-3 8B (dense) | 5.76 |
> | Magnitude | 2.43e+03 |
> | SparseGPT | 18.38 |
> | Wanda | 21.34 |
> | Pruner-Zero | **15.21** |
> | MaskPro | 16.70 |
> | **HyperPrune** | _15.63_ |
>
> ---
>
> ### **W2. GEMM latency improvements vs. other 2:4 methods**
> **Response:**
> We appreciate the reviewer’s comment. On A100 hardware, 2:4 speedups mainly arise from Ampere’s native support for semi-structured sparsity. Consequently, methods that produce valid 2:4 masks typically achieve similar kernel-level acceleration. Our focus is therefore not on kernel benchmarking, but on improving the *quality–efficiency trade-off*: HyperPrune delivers lower perplexity than prior approaches (Table 1) while requiring far less memory and training time (see Q1).
>
> The reported **1.43× end-to-end speedup** represents the expected hardware-aligned benefit of valid 2:4 sparsity. HyperPrune’s main contribution is enabling high-quality mask learning at substantially lower cost.
>
> ---
>
> ### **W3. Drawbacks of hypernetworks (stability, sensitivity, overfitting)**
> **Response:**
> Thank you for raising these points. HyperPrune incorporates two mechanisms that specifically enhance stability:
> (i) **continual pruning** (Sec. 4.2), which enforces cross-layer consistency and prevents interference during sequential pruning, and
> (ii) **sparse-transfer initialization** (Appendix C), which provides a strong starting point and stabilizes optimization.
>
> Our ablations support this: Fig. 4 shows continual pruning improves convergence, and the table below demonstrates that sparse-transfer initialization yields better perplexity and consistency. Overall, we find HyperPrune stable and reproducible across runs. These clarifications will be added to the revised version.
>
> | **Method** | **7B** | **13B** |
> |------------|--------|---------|
> | HyperPrune + SparseGPT prior | **10.11** | **7.60** |
> | HyperPrune + Wanda prior | 10.28 | 7.79 |
> | HyperPrune (no prior) | 10.75 | 8.04 |

---

### Meta-Review · Area_Chair_4MnR · 2026-01-05

**Summary:**

The paper proposes a framework utilizing a shared, context-aware hypernetwork to learn N:M semi-structured sparsity masks for LLMs named HyperPrune. The method effectively bridges the gap between fast heuristics and expensive optimization methods by offering a scalable solution capable of pruning 70B models on a single A100. The authors successfully addressed key concerns during the rebuttal, providing clear evidence of computational efficiency compared to baselines (e.g., MaskLLM) and demonstrating generalization capabilities on modern architectures like LLaMA-3. The AC recommends Accept (Poster) based on the positive consensus among reviewers (Scores: 8, 6, 6, 6).

**Reviewer Concerns:**

**Addressed (Resolved)**:
1. **Generalization to Modern Architectures other than LLaMA-2 (Reviewers c5Ha, 2WfL, 1Unw)**: Reviewers criticized the reliance on the older LLaMA-2 suite. The authors provided new results on LLaMA-3 8B, showing HyperPrune outperforms Wanda, SparseGPT, and MaskPro, and is competitive with Pruner-Zero.
2. **Training Cost & Efficiency (Reviewers c5Ha, 1Unw)**: Reviewers requested a specific breakdown of computational costs. The authors provided a table showing HyperPrune requires significantly less memory and GPU hours (e.g., 15 hours/21GB for 13B) compared to MaskPro (22h/68GB) and MaskLLM (~2300h/630GB).
3. **Hypernetwork Design (Reviewers 2WfL)**: Reviewers asked for justification of the hypernetwork architecture (4-layer FFN) and embedding dimensions. The authors clarified their grid search process (testing widths 128-1024, depths 1-3).
4. **Baselines (Reviewer 1Unw)**: The reviewer requested comparisons to Pruner-Zero, AlphaPruning, and OWL. The authors noted Pruner-Zero was already in Table 1 and explained that AlphaPruning/OWL target mixed/unstructured sparsity, which is incompatible with the uniform N:M hardware constraints addressed in this paper.

**Outstanding (Partially Resolved)**:
1. **Performance at Scale (Reviewer 2WfL)**: The reviewer noted that on the 70B model, HyperPrune lags behind Pruner-Zero (4.87 vs 5.13 PPL). The authors acknowledged this but highlighted that they still outperform Wanda and SparseGPT while being much cheaper to train than full optimization methods.

**Reviewer Scores:**

- Reviewer c5Ha: 6.
- Reviewer 2WfL: 6.
- Reviewer 8US3: 8.
- Reviewer 1Unw: 4 -> 6 after rebuttal.

---

### Decision · Program_Chairs · 2026-01-26

Accept (Poster)